# Derivatives Incorporating Acridine, Pyrrole, and Thiazolidine Rings as Promising Antitumor Agents

**DOI:** 10.3390/molecules28186616

**Published:** 2023-09-14

**Authors:** Monika Garberová, Ivan Potočňák, Monika Tvrdoňová, Monika Majirská, Martina Bago-Pilátová, Slávka Bekešová, Andrej Kováč, Peter Takáč, Krutika Khiratkar, Zuzana Kudličková, Ján Elečko, Mária Vilková

**Affiliations:** 1Institute of Chemistry, Faculty of Science, Pavol Jozef Šafárik University, Moyzesova 11, 040 01 Košice, Slovakia; monika.garberova@student.upjs.sk (M.G.); ivan.potocnak@upjs.sk (I.P.); monika.tvrdonova@upjs.sk (M.T.); jan.elecko@upjs.sk (J.E.); 2Department of Pharmacology, Faculty of Medicine, Pavol Jozef Šafárik University, Trieda SNP 1, 040 01 Košice, Slovakia; monika.majirska@student.upjs.sk (M.M.); martina.pilatova@upjs.sk (M.B.-P.); 3Thermo Fisher Scientific, Mlynské Nivy 5, 821 09 Bratislava, Slovakia; slavka.bekesova@thermofisher.com; 4Department of Pharmacology and Toxicology, University of Veterinary Medicine and Pharmacy, Komenského 73, 041 81 Košice, Slovakia; andrej.kovac@uvlf.sk (A.K.); peter.takac@uvlf.sk (P.T.); 5Institute of Neuroimmunology, Slovak Academy of Sciences, Dúbravská Cesta 9, 845 10 Bratislava, Slovakia; krutika.khiratkar@savba.sk

**Keywords:** acridine, pyrrole, thiazolidine, NMR spectroscopy, X-ray analysis, IR spectroscopy, HR MS spectroscopy, antiproliferative activity

## Abstract

Derivatives combining acridine, pyrrole, and thiazolidine rings have emerged as promising candidates in the field of antitumor drug discovery. This paper aims to highlight the importance of these three structural motifs in developing potent and selective anticancer agents. The integration of these rings within a single molecule offers the potential for synergistic effects, targeting multiple pathways involved in tumor growth and progression. Spiro derivatives were efficiently synthesized in a two-step process starting from isothiocyanates and 2-cyanoacetohydrazide. The thiourea side chain in spiro derivatives was utilized as a key component for the construction of the thiazolidine-4-one ring through regioselective reactions with bifunctional reagents, namely methyl-bromoacetate, dietyl-acetylenedicarboxylate, ethyl-2-bromopropionate, and ethyl-2-bromovalerate. These reactions resulted in the formation of a single regioisomeric product for each derivative. Advanced spectroscopic techniques, including 1D and 2D NMR, FT-IR, HRMS, and single-crystal analysis, were employed to meticulously characterize the chemical structures of the synthesized derivatives. Furthermore, the influence of these derivatives on the metabolic activity of various cancer cell lines was assessed, with IC_50_ values determined via MTT assays. Notably, derivatives containing ester functional groups exhibited exceptional activity against all tested cancer cell lines, boasting IC_50_ values below 10 μM. Particularly striking were the spiro derivatives with methoxy groups at position 3 and nitro groups at position 4 of the phenyl ring. These compounds displayed remarkable selectivity and exhibited heightened activity against HCT-116 and Jurkat cell lines. Additionally, 4-oxo-1,3-thiazolidin-2-ylidene derivatives demonstrated a significant activity against MCF-7 and HCT-116 cancer cell lines.

## 1. Introduction

Cancer remains a major global health concern, necessitating the continuous exploration of novel therapeutic agents. Derivatives incorporating acridine, pyrrole, and thiazolidine rings have gained attention as potential antitumor agents due to their unique chemical properties and diverse biological activities. The integration of these three rings into a single molecule provides a multi-targeted approach, addressing key hallmarks of cancer and offering the potential for improved efficacy and reduced side effects.

The acridine ring is known for its ability to intercalate into DNA, leading to the disruption of DNA replication and transcription [1,2]. This property imparts cytotoxicity and anti-proliferative effects on cancer cells. Acridine derivatives have shown potential in inducing apoptosis [3], inhibiting tumor cell growth [4], and interfering with DNA repair mechanisms [5,6]. These compounds have demonstrated activity against a variety of cancers, including breast [7], lung [8], colon [9], and leukemia [10]. Furthermore, acridine-based compounds have the potential to overcome multidrug resistance mechanisms, making them valuable candidates for combination therapies.

Pyrrole derivatives possess anti-inflammatory properties [11], which can be beneficial in the context of cancer. Chronic inflammation is closely associated with tumor development and progression [12]. By targeting inflammatory pathways, pyrrole-based compounds have the potential to modulate the tumor microenvironment and inhibit cancer cell proliferation [13]. Additionally, these derivatives exhibit antioxidant activity, protecting cells from oxidative stress-induced damage [14]. The anti-inflammatory and antioxidant properties of pyrrole derivatives contribute to their potential as antitumor agents [12,14].

Thiazolidine derivatives have demonstrated antidiabetic effects by improving insulin sensitivity and reducing hyperglycemia [15]. In the context of cancer, dysregulated glucose metabolism is a hallmark of tumor cells [16]. Thiazolidine derivatives can potentially target cancer cells with altered glucose utilization, leading to metabolic stress and cell death [17]. Moreover, these compounds exhibit anti-inflammatory properties [18,19], which can suppress tumor-promoting inflammation and contribute to their anticancer activity [20,21].

The integration of acridine, pyrrole, and thiazolidine rings within a single molecule offers the potential for synergistic effects, targeting multiple cellular pathways involved in tumor growth and survival (Figure 1).

In this ongoing long-term exploration of antiproliferative agents, our laboratory designed, synthesized, and rigorously evaluated a series of derivatives incorporating acridine, pyrrole, and thiazolidine rings. This endeavor aims to broaden chemical diversity and significantly enhance our understanding of structure–activity relationship (SAR) profiles.

## 2. Results and Discussion

### 2.1. Chemistry and NMR Spectroscopy

The synthesis of derivatives **6a**–**e** involved the reaction of the respective cyanoacetohydrazides **3a**–**e** with acridine-9-carbaldehyde (**4**) [22], as depicted in Figure 1. The first step involved the synthesis of hydrazides **3a**–**e**, which were prepared from commercially available isothiocyanates **1a**–**e** and 2-cyanoacetohydrazide (**2**) in ethanol at room temperature. The second step entailed the synthesis of spiro derivatives **6a**–**e**. Acridine-9-carbaldehyde (**4**) was allowed to react with the appropriate hydrazides **3a**–**e**, catalyzed by sodium acetate in dry ethanol at room temperature. The compounds were subsequently purified by crystallization from acetone. The final yields of purified spiro compounds **6a**–**e**, expressed in terms of starting acridine-9-carbaldehyde (**4**), ranged from 59 to 74%. Intermediates **6a**–**e** were subjected to modifications using bifunctional reagents MBA, DEAD, EBP, and EBV. Almost all reactions were conducted in the presence of Et_3_N in ethanol, except for the reaction of **6e** with MBA, which was realized in the presence of anhydrous sodium acetate at 70 °C in ethanol. The target molecules **7**–**10** were obtained with moderate yields ranging from 38 to 83% (Figure 1). The synthesized compounds underwent further analysis using high-field NMR techniques.

Preliminary measurements of the ^1^H NMR spectra of spiro derivatives **6a**–**e** in DMSO-*d*_6_ showed only broadened signals. This prompted us to obtain NMR spectra in acetone-*d*_6_, where the signals exhibited significantly less broadening. This allowed us to measure the ^13^C and 2D NMR spectra for all spiro derivatives **6a**–**e**. Based on 2D NMR data, including gCOSY, NOESY, gHSQC, and gHMBC, all chemical shifts and multiplicities were reliably and unambiguously assigned. Other solvents, such as methanol-*d*_4_, acetonitrile-*d*_3_, and benzene-*d*_6_, also reduced line broadening (Figure 2), but due to the poor solubility of spiro derivatives **6a**–**e**, it was not possible to obtain high-quality ^13^C and 2D spectra.

The determination of the structure relied on crucial observations. The presence of a quaternary sp^3^-hybridized carbon, C-9′, at approximately 70 ppm indicated spiro formation, supported by its chemical shift several ppm downfield compared to the typical spiro carbon of acridine [23]. The signals of C-4′a,10′a carbons around 140 ppm also confirmed the spirocyclic structure. A nitrile carbon, C-6, resonated at approximately 112 ppm, and a thiocarbonyl group was evident from the resonance of the C=S carbon at around 182 ppm and the C=O carbon at about 164 ppm. The resonance of the methine proton H-3 at about 8.50 ppm showed strong deshielding due to acridine’s magnetic anisotropy, supporting the structure **6** over structure **5**, with further validation using the HMBC method. The ^1^H NMR spectra of **6** revealed three distinct one-proton NH signals. The NH-10′ proton assignment was unambiguous, evidenced by its NOESY correlations to H-4′,5′ acridine protons. NH-10 was assigned through NOESY cross-peaks with signals from phenyl ortho protons. However, NH-8 did not display any NOESY cross-peaks. COSY and NOESY correlations resolved almost all acridine proton signals (H-2′,7′ to H-4′,5′) and the HSQC spectrum determined the attached carbon signals (C-2′,7′ to C-4′,5′). HMBC spectra were used to assign the carbons C-4′a,10′a, C-8′a,9′a, and C-9′ based on their correlations with specific protons (Figure 3).

The NMR spectra of derivatives **6a**–**e** in acetone-*d*_6_ exhibit a distinctive characteristic: the presence of doubled signals for the phenyl protons and carbons, while the protons bonded to nitrogen atoms remain unaffected by this doubling phenomenon.

The spiro derivatives **6a**–**e** feature a C=S thiocarbonyl group and two NH labile amide protons, which introduce a propensity for a range of intramolecular transformations in a solution [24,25]. These transformations lead to distinct chemical environments for specific proton and carbon atoms, resulting in variations in chemical shifts for the same proton and carbon nucleus, thus manifesting as signal doubling. The dynamic properties stemming from these intramolecular rearrangements are frequently discernible in the ^1^H and ^13^C NMR spectra [26]. Solvent effects also present another potential factor for the observed signal doubling. The selection of solvent, as in the case of acetone-*d*_6_, can cause the presence of different solvated forms or complexes of the molecule, thereby contributing to signal splitting in the NMR spectrum.

For the instance of compound **6c**, the ^1^H NMR spectrum in acetone-*d*_6_ unequivocally discloses the presence of signals from H-2″ at 6.88 ppm (t, *J* 2.3 Hz) and 6.86 (t, *J* 2.3 Hz), with integral intensities in a 60:40 ratio. Some of the remaining phenyl protons experienced partial or complete overlap with one another. The doubling of certain resonances in the ^13^C NMR spectrum of **6c** was evident (Figure 4). Analogously, the doubling of specific resonances in both the ^1^H and ^13^C NMR spectra was a consistent observation across all spiro derivatives **6** (Appendix A). It is notable that the ^1^H and ^13^C NMR spectra of derivative **6b** in acetonitrile-*d*_3_ exhibited no instances of doubled NMR signals (Appendix A).

We postulated that the observed broadening of signals for H-1″,8″ and C-1″,8″ arises from the dynamic processes described earlier as well as solvent effects. These two factors contribute to the distinct chemical environments experienced by these atoms.

The incorporation of the thiourea chain within the molecular structure of spiro derivatives **6a**–**e** facilitated the formation of thiazolidin-4-one rings through reactions with bifunctional reagents, namely methyl bromoacetate (MBA), diethyl acetylenedicarboxylate (DEAD), ethyl 2-bromopropionate (EBP), and ethyl 2-bromovalerate (EBV) (Figure 1). These reactions involved two-step processes, wherein the bromine of reagent (in the case of MBA, EBP, and EBV) was displaced by the sulfur of the spiro compound **6**, leading to the formation of S-substituted intermediates. The subsequent nucleophilic attack by one of the nitrogen atoms on the carboxylate carbon triggered the elimination of the methoxide or ethoxide ions, leading to cyclization. Despite the potential for obtaining two regioisomeric products, the treatment of the spiro compounds **6a**–**e** with MBA, DEAD, EBP, and EBV yielded only a single product **7**–**10**, as evidenced by the ^1^H NMR spectra of the reaction mixtures. This outcome was consistent with previous findings in similar systems, where MBA, bromoacetyl bromide, or DEAD yielded exclusively one regioisomer, as observed with (acridin-9-yl)methylthioureas [27] and 3-[(*E*)-[(acridin-9-yl)methylidene]amino]-1-substituted thioureas [28]. Our results demonstrate the simple regioselective formation of the novel spiro acridine thiazolidinones **7**–**10**, the structures of which were confirmed by 1D and 2D experiments (Figure 5). This indicates that the regioselectivity could be attributed to the higher nucleophilicity of N-10 nitrogen in its amino form compared to N-8 [29].

Reactions with MBA, EBP, and EBV were conducted at room temperature and catalyzed by Et_3_N. The initial yellow reaction mixtures promptly turned orange upon Et_3_N addition, indicating rapid substitution reactions between the substrate and the reagent, accompanied by the release of bromine. Notably, the reactions of spiro derivatives **6a**–**e** with MBA and EBP exhibited the fastest reaction times, ranging from 2 to 3 h (Appendix A). However, the reaction involving the 4-nitroderivative **6e** with MBA and triethylamine proved unsuccessful. Addressing this, we modified the reaction conditions by employing sodium as the base. This approach yielded product **7e** after 3 h of reflux, albeit with a modest yield of 41% compared to other reactions. In contrast, the reactions of spiro compounds **6a**–**e** with DEAD proceeded spontaneously without catalysis, under reflux conditions in ethanol. Interestingly, these reactions with DEAD demanded a significantly longer reaction time, approximately 7 h, which remained largely unaffected by varying substituents on the phenyl nucleus (Appendix A). Conversely, reactions with EBV necessitated even lengthier periods, ranging from 24 to 48 h. In an attempt to expedite these reactions, we explored the use of sodium acetate with EBV under reflux. However, the reaction mixtures rapidly turned dark brown and yielded complex substances that defied separation, precluding definitive structural determination (Figure 1).

The geometry of the C9=N8 double bond was not determined using NMR experiments due to sparse proton data. However, based on previous knowledge and experience [28], it was assumed that the reactions produce *Z*_C9=N8_-isomers, as the *Z*-configuration is sterically more favorable. The geometric configuration of the C12=C14 double bond was resolved based on heteronuclear coupling constants extracted from EXSIDE spectra (Figure 6) [30], indicating that the C12=C14 double bond in isomers **8a**–**e** exists in the *Z*-configuration [28], with coupling constants ^3^*J*_C11/H14_ ranging from 3.54 to 5.11 Hz (Table 1).

### 2.2. Chiral HPLC Separation

To resolve the enantiomeric derivatives **9** and **10**, chiral HPLC was employed. The results are summarized in Table 2, and Figure 7 illustrates a representative HPLC chromatogram. In most cases, two peaks of equal intensity were observed, indicating the presence of a racemic mixture comprising two enantiomers. However, enantioseparation could not be achieved for derivatives **9b** and **10b** due to their insolubility in common HPLC solvents compatible with the used chiral column, thus hindering the determination of their enantiomeric purity. Baseline enantioseparation and successful chiral recognition were achieved for derivatives **10a** and **10c**–**d**, using the Chiralpak IA column and *n*-hexane/acetone as the mobile phase. On the other hand, the enantiomeric separation of **9a** was unsuccessful due to signal overlap, though the peak’s shape suggested the presence of both enantiomers. Despite attempts to optimize conditions by varying the mobile phase, temperature, or flow rate, the signal resolution could not be improved. Based on the similarity with other measured samples, we infer that they also exist as racemic mixtures. While some samples exhibited a poorer signal resolution, all the measured compounds were considered racemic mixtures.

### 2.3. Infrared Spectroscopy

The analysis of infrared spectra provided the confirmation of the structure of derivatives **6**–**10**, as evidenced by the presence of characteristic vibrational bands from the functional groups present in the studied molecules. Figure 8 illustrates the spectral difference between the starting spiro derivative **6c** and products **7c**–**10c**. In the FT-IR spectra of spiro derivatives **6a**–**d**, the appearance of three weak broad absorption bands above 3200 cm^−1^ indicated the presence of three NH groups from acridine and thiourea moieties. However, in the spectra of **7c**–**10c**, the stretching thiourea vibration disappeared after the formation of the thiazolidine-4-one ring, and only one amino absorption band was observed, confirming cyclization involving the NH groups. In the 2800–3080 cm^−1^ region, weak aromatic and aliphatic CH stretches were present in all compounds. A very weak absorption peak at around 2230 cm^−1^ was detected, assigned to the stretching vibration of the cyano group. Moreover, two C=O vibration bands in the range of 1686–1741 cm^−1^ were evident for derivatives **7**–**10**, while only one band was present in the **6a**–**e** spectra, indicating the formation of a new carbonyl group. A new strong characteristic band at around 1580 cm^−1^, belonging to the stretching vibration of the C=N double bond, was observed in the double bonds’ spectral range of molecules **7**–**10**. Numerous typical stretching aromatic C=C vibrational bands were also observed in this region. Absorption bands at around 1250 cm^–1^ could be attributed to the stretching vibration of C-N single bonds. Additionally, under the ring deformation vibrations at around 750 cm^−1^, stretching C–S vibrations of the thiazolidin-4-one ring were observed at around 670–690 cm^−1^. The infrared spectra of all synthesized compounds **6**–**10** can be found in ESI (Appendix A).

### 2.4. X-ray Analysis

The structure of **6a** was initially determined from NMR data and further confirmed by single-crystal X-ray structure analysis. The crystal structure of **6a**·**AC** (Appendix B) revealed that it crystalizes in the monoclinic space group *P*2_1_/*c*. In the solid state, it comprises a 3-{4′-cyano-5′-oxo-1′,5′-dihydro-10*H*-spiro[acridine-9,2′-pyrrol]-1′-yl]-1-phenyltiourea molecule (**6a**) and a solvated acetone molecule (**AC**) connected to **6a** through a hydrogen bond. Figure 9 illustrates that the **6a** molecule consists of three linked rings, with none of them being coplanar with each other. The planar phenyl and pyrrole rings are joined by a thiourea linker, forming an angle of 60.92(7)°. In accordance with our previous results [23], the dihydroacridine moiety in **6a** is nearly planar. On the other hand, in a related 1-[(*E*)-[(4″-bromophenyl)methylidene]amino]-5-oxo-1,5-dihydro-10′*H*-spiro[acridine-9′,2-pyrrole]-4-carbonitrile compound [31], the dihydroacridine moiety is slightly parabolically curved as was also observed in some structures of dihydroacridines [32]. The mean plane of the dihydroacridine moiety in **6a** is in a nearly orthogonal position towards the pyrrole ring (87.52(4)°), partially due to the steric repulsion between the two mentioned systems and partially due to the *sp*^3^ hybridized C2 atom that is shared by both systems and links them. Finally, the angle of the mean planes passing through the phenyl and acridine units is 74.55(6)°. The bond lengths and angles (Table 3) characterizing the geometry of the dihydroacridine, phenyl, and pyrrole moieties in **6a** are similar to those found in the earlier investigated compounds with the corresponding rings [32,33]. Similarly, the bond lengths of the C=O and C–C≡N bonds and the bonds within the thiourea linker and the acetone molecule are within the usual lengths for such bonds [34].

The two adjacent molecules of **6a** form a dimer linked by hydrogen bonds N24–H24···S11^i^ and N24^i^–H24^i^···S11 (i = −x, 1 − y, 1 − z), and this dimer can be characterized using the graph set descriptor $R_{2}^{2}(18)$ [35]. Additionally, a pair of solvated acetone molecules is attached to this dimer through hydrogen bonds involving N12 atoms as donors and O35 atoms as acceptors (Table 4, Figure 10) (the graph set descriptor is $D_{1}^{1}(2)$), forming **6a·AC**. However, at room temperature, these molecules gradually detach, leading to the decomposition of the crystal and subsequently affecting the data quality for the structure determination. Despite the presence of aromatic rings in **6a**, no π–π interactions were observed between the rings.

### 2.5. HR MS Spectra

The mass spectral fragmentation of the prepared heterocyclic compounds containing acridine, pyrrole, and thiazolidine rings was thoroughly investigated. In the mass spectra of all compounds **6**–**10**, a prominent molecular ion peak (base peak) corresponding to their respective molecular formulas was observed, reaffirming their structures with mass errors of less than 3 ppm (Appendix A). The mass spectra of compounds **6**–**10** are entirely consistent with the assigned structures. Moreover, the precursor ions were systematically fragmented using distinct fragmentation techniques (CID and HCD) at various collision energies. These experiments resulted in a comprehensive collection of fragmentation spectra at each fragmentation level, spanning from MS^2^ to MS^4^, allowing for the construction of a comprehensive spectral tree of information (Figure 11 and Figure 12).

During fragmentation, it was notable that the different substituents on the phenyl ring in compounds **6** did not affect the formation of the two characteristic fragmentation pathways, despite their varying electronegativities. In all derivatives **6a**–**d**, two fragment ions with identical *m*/*z* values emerged after MS^2^ fragmentation: 289.10839 (100%) and 331.06481 (15%) (Figure 11, paths I and II). The sole observed variation was in **6e** (NO_2_, electron-withdrawing group), where a slightly reduced intensity (by 50%) in the formation of the fragment ion with *m*/*z* 331.06481 (6) was observed compared to the electron-donating substituents. The observation of fragment ions from the protonated compounds **6a**–**d** revealed that the C=S group significantly enhances the tendency for elimination. As illustrated in Figure 11, these fragmentations predominantly occurred at the HN–(C=S)–NH bond, involving H-rearrangement. This likely created an occasion for a preferential intramolecular 1,2-elimination (or intramolecular β-elimination), subsequently forming a new π-bond [36], leading to the neutral loss of phenyl fragments.

For compounds **7**–**10**, which contain thiazolidine-4-one rings, two dominant fragments were also observed in the MS^2^ spectra. However, these fragments followed distinct characteristic fragmentation pathways compared to compounds **6**. In these cases, elimination still occurred preferentially but resulted in a ring contraction product (Figure 12, *m*/*z* 231.09617, 100%, path I, ring contraction).

A new π-bond was formed between C=N with the subsequent neutral loss of the entire system containing the thiazolidine-4-one and phenyl rings. The formation of the fragment ion with *m*/*z* 272.08184 (80%) was presumed to involve a transition state where the vacant orbital, being formed on N-1 during the heterolytic cleavage of the N1–N8 bond, was stabilized through interaction with the lone pair of acridine nitrogen N-10′ (Figure 12, path IIa) or oxygen in pyrrole (path IIb). As this ion was considerably intense for all compounds **7**–**10**, it is likely that the formation of ion *m*/*z* 272.0818 required a relatively low activation barrier, making it a kinetically favored process.

### 2.6. Biological Activity

The impact of acridine spiro derivatives **6**–**10** (Figure 13) on the metabolic activity of various tumor cell lines HCT-116 (human colorectal carcinoma), Jurkat (human leukemia cells), HeLa (human cervical adenocarcinoma), MDA-MB-231, and MCF-7 (human breast adenocarcinoma), and normal cell lines COS-7 (monkey kidney cells) and BJ-5ta (skin fibroblasts) was investigated using the MTT assay. This assay measures the conversion of yellow 3-(4,5-dimethylthiazol-2-yl)-2,5-diphenyltetrazolium bromide (MTT) to insoluble purple formazan by metabolically active cells [37]. The antiproliferative activities of the tested compounds were determined based on the results from the MTT assay, and the inhibitory concentration values (IC_50_) are presented in Table 5.

Among the derivatives investigated, those containing the thiosemicarbazide group in the structure, spiro derivatives **6**, demonstrated a relatively effective activity. Compounds **6a**,**c**–**e** showed activity against HCT-116 colorectal carcinoma cells with IC_50_ values ranging from 7.00 to 19.57 μM, and against Jurkat leukemia cells with IC_50_ values ranging from 6.47 to 31.75 μM. Moreover, unlike the derivatives **8**, they exhibited low activity against normal cells COS-7 and BJ-5ta. Spiro compound **6c**, with a 3-methoxyphenylthiosemicarbazide chain, exhibited the highest activity, demonstrating selectivity against HCT-116 and Jurkat tumor cells with a selectivity index exceeding 10 compared to the non-tumor cells COS-7 and BJ-5ta (Table 5, Figure 14).

Derivatives **7** showed moderate activity, as they were not substituted at the position 5 of the thiazolidine-4-one ring, unlike the other investigated compounds. They displayed the highest activity against the breast cancer cell line MCF-7 and the colorectal carcinoma cell line HCT-116. In this case as well, the activity of derivatives **7** was higher against tumor cells compared to normal cells (Table 5).

The study revealed that the derivatives containing methyl (**9**) (Table 5) and propyl (**10**) (Table 5) groups on the thiazolidine-4-one ring were the least active. The presence of a polar ester group attached via the methylene group to the thiazolidine-4-one (**8**) was found to be crucial for biological activity. All compounds with this substitution **8a**–**e** exhibited excellent activity against tumor cell lines (IC_50_ less than 17 μM). However, a notable drawback was their similar toxicity towards normal cell lines COS-7 and BJ-5ta, limiting their potential use in anticancer therapy (Table 5).

The comprehensive SAR investigation sheds light on the connections between the compounds’ unit properties and their effectiveness in inhibiting cancer cell proliferation. Notably, the presence of a methoxy group at position 3, a nitro group at position 4 on the phenyl ring, a thiourea linker, and the inclusion of the (1,3-thiazolidin-5-ylidene)acetate unit demonstrated significant enhancements in antiproliferative activities, resulting in a remarkable increase in their inhibitory effects (Figure 15).

To comprehensively evaluate the cytotoxic effects of the investigated compounds **6c**, **6e**, and **8a**, **8c**, we conducted tests on primary endothelial cells. This additional assessment served multiple purposes in our study. Firstly, it aimed to assess the selectivity of these compounds, particularly in comparison to their effects on both tumor and normal cell lines. The goal was to identify the compounds that exhibit a desirable level of selectivity, showing potential for targeted therapy against cancer cells. Secondly, by examining their impact on endothelial cells, we aimed to explore potential off-target effects, as endothelial cells are crucial components of blood vessels and play a pivotal role in angiogenesis—a process intricately linked to tumor growth [38]. Lastly, these tests allowed for a broader safety assessment, beyond the scope of tumor and normal cell lines, providing insights into the compounds’ implications for vascular health and their regulatory role in angiogenesis. As shown in Table 6, the derivative **8a** exerted the most intensive cytotoxic activity and, on the other hand, the compound **6e** reduced the survival of endothelial cells the least.

## 3. Materials and Methods

### 3.1. General

All reagents were used as supplied without prior purification. The progression of the reaction was monitored by analytical thin-layer chromatography using TLC sheets ALUGRAM-SIL G/UV254 (Macherey Nagel, Dueren, Germany). Purification by flash chromatography was performed using silica gel (60 Å, 230–400 mesh, Merck, Darmstadt, Germany) with the indicated eluent.

### 3.2. Melting Point Determination

The melting points of the synthesized derivatives **6**–**10** were determined using a Stuart^TM^ melting point apparatus SMP10. However, during the melting point determination, it was observed that the compound underwent decomposition before reaching its true melting point. Several attempts were made to obtain a reliable melting point, including varying the heating rate and using different apparatuses, but the compounds continued to decompose. As a result, the definitive melting points could not be obtained for these derivatives, and only decomposition points (d) were observed.

### 3.3. NMR Spectroscopy

NMR spectra were acquired using a Varian VNMRS spectrometer operating at 599.87 MHz for ^1^H, 150.84 MHz for ^13^C, and 60.79 MHz for ^15^N. These experiments were conducted at a temperature of 299.15 K, and a 5 mm inverse-detection H-X probe with a z-gradient coil was used. Pulse programs from the Varian sequence library were employed. Chemical shifts (*δ* in ppm) were referenced to internal solvent standards: DMSO-*d*_6_ at 39.5 ppm, acetone-*d*_6_ at 29.8 ppm, and CD_3_OD 49.0 ppm for ^13^C, while the partially deuterated signals of DMSO-*d*_5_ at 2.5 ppm, acetone-d_5_ at 2.05 ppm, and CHD_2_OD at 3.31 ppm were used for ^1^H referencing. For ^15^N NMR, external nitromethane served as the reference at 0.0 ppm. MestReNova v. 14.2.1 (Mestrelab Research, Santiago de Compostela, Spain) was utilized for NMR spectra processing and analysis.

### 3.4. X-ray Data Collection and Structure Refinement

The data collection for **6a·AC** (Appendix B) was performed using an Oxford Diffraction Xcalibur2 diffractometer equipped with a Sapphire2 CCD detector. CrysAlisPro software, version 1.0.43 was utilized for data collection, cell refinement, data reduction, and absorption correction. The structure was solved with SHELXT [39] and refined using SHELXL-2018 [40] in the WinGX program suite [41]. Hydrogen atoms bonded to carbon atoms were placed at calculated positions and refined by riding on their parent atoms. Geometric analysis and hydrogen bonds analysis were conducted using SHELXL-2018, while PLATON [42] was used to analyze the π–π interaction and DIAMOND [43] was employed for molecular graphics. A summary of the crystal data and structure refinement for **6a·AC** is presented in Table 7.

### 3.5. IR Spectroscopy

The infrared spectra of the synthesized compounds were recorded using an Nicolete 6700 FT-IR spectrophotometer (Thermo Fisher Scientific, Waltham, MA, USA) in the range of 400–4000 cm^−1^ with 64 repetitions for each spectrum, employing the ATR (attenuated total reflectance) technique. All acquired data were analyzed using Omnic 8.2.0.387 (2010) software. The FT-IR spectra were examined to confirm the structure of each new compound and identify the presence of specific functional groups.

### 3.6. Chiral HPLC Conditions

Analytical high-performance liquid chromatography (HPLC) experiments were conducted using a Shimadzu HPLC system, which included a CBM-20A system controller, LC-20AD pump, CTO-20AC column oven, and RID-10A detector. The analysis was conducted with a flow rate of 0.5 mL/min and a column temperature maintained at 25 °C. For the isocratic analysis, a Chiralpak IA column from Daicel was employed, with the mobile phase consisting of *n*-hexane and acetone in varying percentages. Prior to usage, HPLC-grade solvents (*n*-hexane, acetone) were degassed. Sample solutions were prepared by dissolving the compounds in the appropriate mobile phase, resulting in a concentration of approximately 0.5 mg/mL, followed by injection into the system. Data acquisition and instrument control were expertly managed through the LabSolutions Lite software, version 5.52.

### 3.7. HR-MS

The solid samples were dissolved in methanol and then diluted to a final concentration ranging from 1 to 5 µg/mL using methanol containing 0.5% formic acid and 5 mM ammonium formate. The samples were injected using a nanoelectrospray robot, TriVersa NanoMate^®^ (Advion, Ithaca, NY, USA). The volume of the sample aspired into the tip for a single injection was 20 µL, and the maximum spraying rate was approximately 220 nL/min. In the positive mode, the gas pressure (N_2_ extruding the sample from the tip) was set to 0.3 psi, and the applied voltage was 1.4 kV. The samples were injected into a mass spectrometer, Orbitrap Fusion^TM^ Lumos^TM^ Tribrid^TM^ (Thermo Fisher Scientific, Waltham, MA, USA). The ion transfer tube temperature was maintained at 275 °C. Full scan (MS^1^) was performed using an Orbitrap detector with a resolution of 120,000, a maximum injection time of 200 ms, and 2 microscans. Subsequent scans (MS^2^ to MS^4^) were also performed with the Orbitrap detector after fragmentation using CID (collision-induced dissociation) with relative energies ranging from 10 to 100 and HCD (higher-energy collisional dissociation) with relative energies ranging from 10 to 200. The collision pressure was 8 × 10^−3^ Torr. The isolation width for all levels of fragmentation was set to 1 unit (minimum). The automatic gain control was typically set between values of 2 × 10^4^ and 5 × 10^5^, depending on the sample concentration. The obtained data were of high quality, as it included high-resolution MS/MS and multi-stage MSn spectra acquired at various collision energies using different fragmentation techniques. The measured data were manually processed using Mass Frontier™ 8.0 software (Thermo Scientific™, Bratislava, Slovakia) within the Curator module. This module employs advanced algorithms to detect incompatibilities between the declared structure precursor and the product MS^n^ fragmentation spectra. These compounds were added to the high-quality mzCloud™ spectral library (https://www.mzcloud.org). For compounds **6**–**10,** the mzCloud™ ID ranged from 11372 to 11404. HR MS data are tabulated in ESI (Appendix A).

### 3.8. Biological Activity

#### 3.8.1. Cell Lines and Culture Conditions

The human cancer cell lines employed in this study were sourced from reputable institutions, including the American Type Culture Collection (ATCC; Manassas, VA, USA) and the European Collection of Authenticated Cell Cultures (ECACC, Salisbury, UK). Specifically, HeLa (93021013, human cervical adenocarcinoma), HCT116 (CCL-247TM, human colorectal carcinoma), and Jurkat (88042803, human leukemic T-cell lymphoma) were cultured in RPMI 1640 medium (Biosera, Kansas City, MO, USA). In contrast, MCF-7 (HTB-22TM, human Caucasian breast adenocarcinoma, ER+), MDA-MB-231 (HTB-26TM, human breast adenocarcinoma, ER−), and COS-7 (CRL-1651TM, kidney fibroblasts) were nurtured in high-glucose Dulbecco’s Modified Eagle Medium (DMEM) supplemented with sodium pyruvate (Biosera). Furthermore, BJ-5ta (CRL-4001TM, human dermal fibroblasts) was cultured in a composite medium of high-glucose DMEM and M199 (4:1) (Biosera), supplemented with Hygromycin B (0.01 mg/mL; Merck, Darmstadt, Germany). All culture media were enriched with a 10% fetal bovine serum (FBS; Gibco, Thermo Scientific, Rockford, IL, USA) and an antibiotic/antimycotic solution (Merck) to ensure optimal growth conditions. The cells were meticulously maintained within a humidified environment containing 5% CO_2_ at 37 °C. Regular screenings for mycoplasma contamination were conducted using Hoechst 33342 staining (Merck), confirming the integrity of the cell lines.

#### 3.8.2. MTT Assay

To assess the antiproliferative impact of the synthesized compounds **6**–**10**, their half-maximal inhibitory concentration (IC_50_) and the MTT colorimetric assay were employed. MDA-MB-231 and MCF-7 cells were seeded into 96-well microplates (SARSTEDT, Nümbrecht, Germany) at a density of 5 × 103 cells per well. After 24 h of cell seeding, various concentrations (100, 50, and 10 µmol/L) of acridine derivatives **6**–**10** were introduced, alongside cisplatin (utilized as a reference) at concentrations of 5, 10, 20, 50, and 100 µmol/L. Subsequent to 24, 48, and 72 h of treatment, the cells in each well underwent incubation with MTT (5 mg/mL, Sigma-Aldrich Chemie, Steinheim, Germany) for 4 h at 37 °C in the dark. During this period, mitochondrial oxidoreductases metabolized MTT, forming insoluble formazan. Following this, 100 µL of SDS (10% sodium decyl sulfate) was added to dissolve the formazan crystals, and the cells were incubated for an additional 12 h. Absorbance at 540 nm was measured using the automated Cytation™ 3 Cell Imaging Multi-Mode Reader (Biotek, Winooski, VT, USA). Each test was performed in triplicate, and three independent experiments were conducted to ensure precision and reliability [37].

#### 3.8.3. Primary Endothelial Cell Isolation

Primary brain endothelial cells were isolated from rats (200–250 g, 6 months old; n = 5 per isolation). Euthanasia was performed by CO_2_ exposure, followed by brain removal. The brainstem and cerebellum were dissected, and the white matter from the midbrain and choroid plexus was excised under sterile conditions. Cortical tissues were meticulously cleaned of meninges and homogenized on ice in a DMEM-F12 medium (PAA Laboratories GmbH, Cölbe, Germany) with 0.1% bovine serum albumin (BSA, Sigma-Aldrich, St. Louis, MO, USA). The homogenate was centrifuged at 800× *g* for 10 min at 4 °C. After aspirating the supernatant, the pellet was resuspended in a pre-warmed digestion mix containing 1 mg/mL collagenase/dispase (Roche Diagnostics, Indianapolis, IN, USA) and 10 μg/mL DNase I (Roche Diagnostics, Indianapolis, IN, USA). The homogenates were then incubated with the pre-prepared digestion mix at 37 °C for 30 min with gentle shaking. The preparation was centrifuged at 800× *g* for 10 min at 4 °C, and the pellets were resuspended in 20% BSA in the medium. The tissues were centrifuged at 1500× *g* for 15 min at 4 °C to obtain a pellet containing microvessels with a fraction of myelin and BSA on top, which was centrifuged again. The microvessels were pooled and resuspended in a pre-warmed digestion mix and incubated for 15 min at 37 °C. The pellet was then centrifuged at 800× *g* for 10 min at 4 °C and washed with a serum-containing DMEM-F12 culture medium. The microvessels were cultured in a DMEM-F12 medium containing 15% plasma-derived serum (PDS, First Link, Birmingham, UK), 2 mM UltraGlutamine (GE Healthcare, Chalfont Saint Giles, UK), BME vitamins (Sigma-Aldrich, St. Louis, MO, USA), heparin (Sigma-Aldrich, St. Louis, MO, USA), and 3 μM puromycin (Sigma-Aldrich, St. Louis, MO, USA) in 6-well tissue culture plates for the following 3 days. After 3 days, the cells were switched to serum-free media. The endothelial cells were ready to use after 7 days.

#### 3.8.4. Endothelial Cell Viability Assay

The endothelial cell viability was assessed through the quantification of adenosine triphosphate (ATP) released from the cells, utilizing the ATP assay as per the manufacturer’s instructions (The Promega CellTiter-Glo™ Luminescent Cell Viability Assay Kit, Madison, WI, USA). Cell viability data were fitted to an inverse sigmoidal curve, and a four-parameter logistic model was employed to calculate the IC_50_. Data analysis was executed using Prism 8.0 software (GraphPad Inc., San Diego, CA, USA).

### 3.9. General Synthetic Procedure for Compounds ***6a**–**e***

To a mixture of acridine-9-carbaldehyde (**4**, 100 mg, 0.48 mmol) and **3a**–**e** (0.48 mmol) in dry ethanol (2 mL), anhydrous sodium acetate was added (4 mg, 0.05 mmol). The reaction mixture was stirred at room temperature under TLC monitoring (*n*Hex:EtOAc, *v/v* 3:1) until the complete consumption of acridine-9-carbaldehyde (**4**). The formed yellow precipitate was filtered off, washed with a small amount of absolute ethanol, and dried. The crude product was crystallized using hot acetone to yield **6a**–**e**.

*3-{4′-Cyano-5′-oxo-1′,5′-dihydro-10H-spiro[acridine-9,2′-pyrrol]-1′-yl}-1-phenylthiourea* (**6a**). Yellow solid; yield: 74% (0.152 g), m. p.: 194–195 °C (d). ^1^H NMR (600 MHz, acetone-*d*_6_): *δ* 9.06 (s, 1H, H-10′), 8.52 (s, 1H, H-3), 8.34 (s, 1H, H-10), 8.31 (s, 1H, H-8), 7.47 (br s, 2H, H-1′,8′), 7.33 (ddt, *J* = 8.3, 7.1, 1.3 Hz, 2H, H-3′,6′), 7.19 (m, 2H, H-2″,6″), 7.14 (m, 2H, H-3″,5″), 7.08 (m, 3H, H-4′,5′,4″), 6.97 (ddd, *J* = 8.1, 7.1, 1.2 Hz, 2H, H-2′,7′). ^13^C NMR (150 MHz, acetone-*d*_6_): *δ* 182.49 (C-9), 182.35 (C-9), 163.80 (C-5), 162.53 (C-3), 162.52 (C-3), 140.53 (C-4′a,10′a), 140.44 (C-4′a,10′a), 139.36 (C-1″), 139.35 (C-1″), 131.30 (C-3′,6′), 129.10 (C-1′,8′), 128.70 (C-3″,5″), 126.24 (C-4″), 126.22 (C-4″), 125.66 (C-2″,6″), 125.54 (C-2″,6″), 121.74 (C-2′,7′), 121.73 (C-2′,7′), 116.56 (C-4′,5′), 116.48 (C-4′,5′), 112.66 (C-8′a,9′a), 112.60 (C-4), 112.60 (C-6), 70.55 (C-9′), 70.54 (C-9′). ^15^N NMR (61 MHz, acetone-*d*_6_): *δ* −284.8 (NH-10′), −261.5 (NH-8), −255.2 (NH-10), −206.8 (N-1). HRMS: *m*/*z* [M + H]^+^ calcd. for C_24_H_17_N_5_OS: 424.12266; found: 424.12300.

*3-{4′-Cyano-5′-oxo-1′,5′-dihydro-10H-spiro[acridine-9,2′-pyrrol]-1′-yl}-1-(4-methoxyphenyl)thiourea* (**6b**). Yellow solid; yield: 67% (0.148 g), m. p.: 185–188 °C (d). ^1^H NMR (600 MHz, acetone-*d*_6_): *δ* 9.04 (s, 1H, H-10′), 8.51 (s, 1H, H-3), 8.24 (s, 1H, H-10), 8.20 (s, 1H, H-8), 7.48 (br s, 2H, H-1′,8′), 7.35 (ddt, *J* = 8.3, 7.1, 1.1 Hz, 2H, H-3′,6′), 7.09 (d, *J* = 7.9 Hz, 2H, H-4′,5′), 6.97 (ddd, *J* = 8.2, 7.1, 1.2 Hz, 2H, H-2′,7′), 6.93 (d, *J* = 9.0 Hz, 2H, H-2″,6″), 6.74 (d, *J* = 9.0 Hz, 2H, H-3″,5″), 3.74 (s, 3H, OCH_3_). ^13^C NMR (150 MHz, acetone-*d*_6_): *δ* 182.86 (C-9), 182.72 (C-9), 163.73 (C-5), 162.46 (C-3), 158.50 (C-4″), 158.59 (C-4″), 140.53 (C-4′a,10′a), 140.44 (C-4′a,10′a), 132.16 (C-1″), 132.04 (C-1″), 131.25 (C-3′,6′), 129.07 (C-1′,8′), 127.62 (C-2″,6″), 127.54 (C-2″,6″), 121.69 (C-2′,7′), 116.55 (C-4′,5′), 116.47 (C-4′,5′), 113.88 (C-3″,5″), 112.68 (C-8′a,9′a), 112.60 (C-4), 112.60 (C-6), 70.50 (C-9′), 55.60 (OCH_3_). ^15^N NMR (61 MHz, acetone-*d*_6_): *δ* −284.9 (N-10′), −262.7 (N-8), −257.1 (N-10), −206.9 (N-1). HRMS: *m*/*z* [M + H]^+^ calcd. for C_25_H_19_N_5_O_2_S: 454.13322; found: 454.13420.

*3-{4′-Cyano-5′-oxo-1′,5′-dihydro-10H-spiro[acridine-9,2′-pyrrol]-1′-yl}-1-(3-methoxyphenyl)thiourea* (**6c**). Yellow solid; yield: 68% (0.150 g), m. p.: 184–186 °C (d). ^1^H NMR (600 MHz, acetone-*d*_6_): *δ* 9.04 (s, 1H, H-10′), 8.53 (s, 1H, H-3), 8.30 (s, 2H, H-8, H-10), 7.47 (br s, 2H, H-1′,8′), 7.34 (ddt, *J* = 8.4, 7.1, 1.3 Hz, 2H, H-3′,6′), 7.08 (m, 3H, H-4′,5′,5″), 6.99 (ddd, *J* = 8.1, 7.1, 1.2 Hz, 2H, H-2′,7′), 6.86 (dt, *J* = 13.7, 2.3 Hz, 1H, H-2″), 6.76 (tdd, *J* = 8.2, 2.0, 0.9 Hz, 1H, H-6″), 6.65 (ddd, *J* = 8.2, 2.0, 0.9 Hz, 1H, H-4″), 3.72 (s, 3H, OCH_3_). ^13^C NMR (150 MHz, acetone-*d*_6_): *δ* 182.06 (C-9), 182.01 (C-9), 163.87 (C-5), 162.57 (C-3), 160.28 (C-3″), 140.52 (C-4′a,10′a), 140.41 (C-4′a,10′a), 131.26 (C-3′,6′), 129.37 (C-5″), 129.06 (C-1′,8′), 121.73 (C-2′,7′), 121.72 (C-2′,7′), 117.59 (C-6″), 117.45 (C-6″), 116.56 (C-4′,5′), 116.48 (C-4′,5′), 112.64 (C-8′a,9′a), 112.60 (C-4), 112.57 (C-6), 112.08 (C-4″), 112.05 (C-4″), 110.81 (C-2″), 110.68 (C-2″), 70.54 (C-9′), 55.63 (OCH_3_). ^15^N NMR (61 MHz, acetone-*d*_6_): *δ* −284.8 (N-10′), −261.1 (N-8), −254.9 (N-10), −206.8 (N-1). HRMS: *m*/*z* [M + H]^+^ calcd. for C_25_H_19_N_5_O_2_S: 454.13322; found: 454.13390.

*3-{4′-Cyano-5′-oxo-1′,5′-dihydro-10H-spiro[acridine-9,2′-pyrrol]-1′-yl}-1-(4-fluorophenyl)thiourea* (**6d**). Yellow solid; yield: 70% (0.149 g), m. p.: 197–199 °C (d). ^1^H NMR (600 MHz, acetone-*d*_6_): *δ* 9.06 (s, 1H, H-10′), 8.53 (s, 1H, H-3), 8.40 (s, 1H, H-10), 8.36 (s, 1H, H-8), 7.47 (br s, 2H, H-1′,8′), 7.34 (ddt, *J* = 8.4, 7.3, 1.3 Hz, 2H, H-3′,6′), 7.09 (m, 4H, H-4′,5′, H-2″,6″), 6.97 (m, 4H, H-2′,7′,3″,5″). ^13^C NMR (150 MHz, acetone-*d*_6_): *δ* 182.87 (C-9), 182.73 (C-9), 163.65 (C-5), 162.49 (C-3), 162.48 (C-3), 161.10 (dd, *J* = 241.5, 3.0 Hz, C-4″), 140.56 (C-4′a,10′a), 140.47 (C-4′a,10′a), 135.5 (dt, *J* = 16.7, 3.0 Hz, C-1″), 131.27 (C-3′,6′), 129.09 (C-1′,8′), 128.06 (d, *J* = 8.3 Hz, C-2″,6″), 127.96 (d, *J* = 8.3 Hz, C-2″,6″), 121.70 (C-2′,7′), 121.69 (C-2′,7′), 116.54 (C-4′,5′), 116.46 (C-4′,5′), 115.29 (d, *J* = 22.8 Hz, C-3″,5″), 112.70 (C-4), 112.70 (C-6), 112.65 (C-8′a,9′a), 70.56 (C-9′), 70.55 (C-9′). ^15^N NMR (61 MHz, acetone-*d*_6_): *δ* −284.7 (N-10′), −261.6 (N-8), −257.5 (N-10), −206.9 (N-1). HRMS: *m*/*z* [M + H]^+^ calcd. for C_24_H_16_FN_5_OS: 442.11324; found: 442.11340.

*3-{4′-Cyano-5′-oxo-1′,5′-dihydro-10H-spiro[acridine-9,2′-pyrrol]-1′-yl}-1-(4-nitrophenyl)thiourea* (**6e**). Yellow solid; yield: 59% (0.133 g), m. p.: 175–179 °C (d). ^1^H NMR (600 MHz, acetone-*d*_6_): *δ* 9.07 (s, 1H, H-10′), 8.91 (s, 1H, H-10), 8.75 (s, 1H, H-8), 8.59 (s, 1H, H-3), 8.08 (d, *J* = 9.2 Hz, 2H, H-3″,5″), 7.66 (d, *J* = 9.2 Hz, 2H, H-2″,6″), 7.28 (br s, 2H, H-3′,6′), 7.06 (br d, *J* = 8.1 Hz, 2H, H-4′,5′), 6.97 (ddd, *J* = 8.2, 7.2, 1.2 Hz, 2H, H-2′,7′). ^13^C NMR (150 MHz, acetone-*d*_6_): *δ* 181.82 (C-9), 163.73 (C-5), 162.62 (C-3), 145.39 (C-1″), 145.28 (C-1′’), 144.85 (C-4″), 140.44 (C-4′a,10′a), 131.32 (C-3′,6′), 129.05 (C-1′,8′), 124.31 (C-3″,5″), 123.96 (C-2″,6″), 123.81 (C-2″,6″), 121.79 (C-2′,7′), 116.51 (C-4′,5′), 116.43 (C-4′,5′), 112.82 (C-8′a,9′a), 112.80 (C-4), 112.57 (C-6), 70.80 (C-9′). ^15^N NMR (61 MHz, acetone-*d*_6_): *δ* −284.5 (N-10′), −257.2 (N-10), −207.0 (N-1). HRMS: *m*/*z* [M + H]^+^ calcd. for C_24_H_16_N_6_O_3_S: 469.10774; found: 469.10790.

### 3.10. General Synthetic Procedure for Compounds ***7a**–**d***

To a suspension of **6a**–**d** (50 mg) in dry ethanol (0.6 mL), methyl-bromoacetate (1.2 equiv) and triethylamine (1.0 equiv) were added. The reaction mixture was stirred at room temperature. After 1 h, another amount of triethylamine (1.0 equiv) was added. The mixture was stirred for another 2–3 h. The progress of the reaction was monitored by TLC (*n*Hex:EtOAc, *v/v* 1:3). The formed yellow precipitate was filtered off, washed with a small amount of absolute ethanol, and dried.

*5′-Oxo-1′-{[(2Z)-4-oxo-3-phenyl-1,3-thiazolidin-2-ylidene]amino}-1′,5′-dihydro-10H-spiro[acridine-9,2′-pyrrole]-4′-carbonitrile* (**7a**). Yellow solid; yield: 79% (0.043 g), m. p.: 241–243 °C (d). ^1^H NMR (600 MHz, DMSO-*d*_6_): *δ* 9.77 (s, 1H, H-10′), 8.69 (s, 1H, H-3), 7.29 (m, 3H, H-3″,4″,5″), 7.26 (ddd, *J* = 8.4, 7.1, 1.4 Hz, 2H, H-3′,6′), 7.05 (m, 2H, H-2″,6″), 7.02 (dd, *J* = 8.2, 1.2 Hz, 2H, H-4′,5′), 7.02 (dd, *J* = 7.9, 1.4 Hz, 2H, H-1′,8′), 6.81 (ddd, *J* = 7.9, 7.1, 1.2 Hz, 2H, H-2′,7′), 4.03 (s, 2H, H-12). ^13^C NMR (151 MHz, DMSO-*d*_6_): *δ* 171.0 (C-11), 167.0 (C-9), 162.0 (C-3), 157.4 (C-5), 139.2 (C-4′a,10′a), 133.8 (C-1″), 129.8 (C-3′,6′), 128.4 (C-3″,5″), 128.2 (C-4″), 127.7 (C-1′,8′), 127.4 (C-2″,6″), 120.0 (C-2′,7′), 114.8 (C-4′,5′), 112.5 (C-6), 112.4 (C-8′a,9′a), 110.9 (C-4), 69.6 (C-9′), 32.7 (C-12). ^15^N NMR (61 MHz, DMSO-*d*_6_): *δ* −284.4 (N-10′), −217.2 (N-10), −186.9 (N-1). HRMS: *m*/*z* [M + H]^+^ calcd. for C_26_H_17_N_5_O_2_S: 464.11757; found: 464.11790.

*1′-{[(2Z)-3-(4-Methoxyphenyl)-4-oxo-1,3-thiazolidin-2-ylidene]amino}-5′-oxo-1′,5′-dihydro-10H-spiro[acridine-9,2′-pyrrole]-4′-carbonitrile* (**7b**). Yellow solid; yield: 66% (0.036 g), m. p.: 207–210 °C (d). ^1^H NMR (600 MHz, DMSO-*d*_6_): δ 9.74 (s, 1H, H-10′), 8.69 (s, 1H, H-3), 7.27 (ddd, *J* = 8.4, 7.2, 1.4 Hz, 2H, H-3′,6′), 7.02 (dd, *J* = 8.2, 1.4 Hz, 2H, H-1′,8′), 7.02 (dd, *J* = 8.2, 1.4 Hz, 2H, H-4′,5′), 6.95 (d, *J* = 9.0 Hz, 2H, H-2″,6″), 6.83 (ddd, *J* = 8.4, 7.2, 1.4 Hz, 2H, H-2′,7′), 6.82 (d, *J* = 9.0 Hz, 2H, H-3″,5″), 4.00 (s, 2H, H-12), 3.73 (s, 3H, OCH_3_). ^13^C NMR (151 MHz, DMSO-*d*_6_): δ 171.1 (C-11), 167.1 (C-9), 162.0 (C-3), 158.5 (C-4″), 157.3 (C-5), 139.1 (C-4′a,10′a), 129.8 (C-3′,6′), 128.5 (C-2″,6″), 127.7 (C-1′,8′), 126.4 (C-1″), 120.0 (C-2′,7′), 114.8 (C-4′,5′), 113.6 (C-3″,5″), 112.5 (C-6), 112.4 (C-8′a,9′a), 111.0 (C-4), 69.6 (C-9′), 55.3 (OCH_3_), 32.6 (C-12). ^15^N NMR (61 MHz, DMSO-*d*_6_): *δ* −283.5 (N-10′), −218.5 (N-10), −186.0 (N-1). HRMS: *m*/*z* [M + H]^+^ calcd. for C_27_H_19_N_5_O_3_S: 494.12814; found: 494.12880.

*1′-{[(2Z)-3-(3-Methoxyphenyl)-4-oxo-1,3-thiazolidin-2-ylidene]amino}-5′-oxo-1′,5′-dihydro-10H-spiro[acridine-9,2′-pyrrole]-4′-carbonitrile* (**7c**). Yellow solid; yield: 83% (0.045 g), m. p.: 217–220 °C (d). ^1^H NMR (600 MHz, DMSO-*d*_6_): *δ* 9.69 (s, 1H, H-10′), 8.65 (s, 1H, H-3), 7.25 (ddd, *J* = 8.4, 7.2, 1.4 Hz, 2H, H-3′,6′), 7.19 (t, *J* = 8.4 Hz, 1H, H-5″), 7.02 (dd, *J* = 8.2, 1.4 Hz, 2H, H-1′,8′), 6.99 (dd, *J* = 8.2, 1.2 Hz, 2H, H-4′,5′), 6.85 (ddd, *J* = 8.4, 2.4, 1.1 Hz, 1H H-4″), 6.81 (ddd, *J* = 8.2, 7.2, 1.2 Hz, 2H, H-2′,7′), 6.62 (m, 2H, H-2″,6″), 4.03 (s, 2H, H-12), 3.66 (s, 3H, OCH_3_). ^13^C NMR (151 MHz, DMSO-*d*_6_): *δ* 171.0 (C-11), 166.9 (C-9), 162.1 (C-3), 159.1 (C-3″), 157.6 (C-5), 139.1 (C-4′a,10′a), 134.8 (C-1″), 129.7 (C-3′,6′), 129.1 (C-5″), 127.7 (C-1′,8′), 119.9 (C-2′,7′), 119.7 (C-6″), 114.9 (C-4′,5′), 114.1 (C-4″), 113.1 (C-2″), 112.5 (C-6), 112.2 (C-8′a,9′a), 110.3 (C-4), 69.6 (C-9′), 55.2 (OCH_3_), 32.7 (C-12). ^15^N NMR (61 MHz, DMSO-*d*_6_): *δ* −283.8 (N-10′), −186.3 (N-1). HRMS: *m*/*z* [M + H]^+^ calcd. for C_27_H_19_N_5_O_3_S: 494.12814; found: 494.12880.

*1′-{[(2Z)-3-(4-Fluorophenyl)-4-oxo-1,3-thiazolidin-2-ylidene]amino}-5′-oxo-1′,5′-dihydro-10H-spiro[acridine-9,2′-pyrrole]-4′-carbonitrile* (**7d**). Yellow solid; yield: 70% (0.038 g), m. p.: 249–252 °C (d). ^1^H NMR (600 MHz, DMSO-*d*_6_): *δ* 9.73 (s, 1H, H-10′), 8.69 (s, 1H, H-3), 7.27 (ddd, *J* = 8.4, 7.1, 1.4 Hz, 2H, H-3′,6′), 7.12 (t, *J* = 8.8 Hz, 2H, H-3″,5″), 7.09 (dd, *J* = 9.0, 5.1 Hz, 2H, H-2″,6″), 7.02 (dd, *J* = 8.0, 1.4 Hz, 2H, H-1′,8′), 7.01 (dd, *J* = 8.4, 1.2 Hz, 2H, H-4′,5′), 6.82 (ddd, *J* = 8.0, 7.1, 1.2 Hz, 2H, H-2′,7′), 4.02 (s, 2H, H-12). ^13^C NMR (151 MHz, DMSO-*d*_6_): *δ* 171.0 (C-11), 166.7 (C-9), 162.0 (C-3), 160.3 (d, *J* = 245.6 Hz, C-4″), 157.5 (C-5), 139.1 (C-4′a,10′a), 130.0 (d, *J* = 2.9 Hz, C-1″), 129.8 (C-3′,6′), 129.7 (d, *J* = 8.9 Hz, C-2″,6″), 127.7 (C-1′,8′), 120.1 (C-2′,7′), 115.4 (d, *J* = 22.9 Hz, H-3″,5″), 114.8 (C-4′,5′), 112.5 (C-6), 112.4 (C-8′a,9′a), 111.0 (C-4), 69.7 (C-9′), 32.9 (C-12). ^15^N NMR (61 MHz, DMSO-*d*_6_): *δ* −283.5 (C-10′), −219.1 (N-10), −186.3 (N-1). HRMS: *m*/*z* [M + H]^+^ calcd. for C_26_H_16_FN_5_O_2_S: 482.10815; found: 482.10870.

### 3.11. Synthetic Procedure for Compound ***7e***

To a suspension of **6e** (50 mg, 0.11 mmol) in dry ethanol (0.6 mL), anhydrous sodium acetate was added (35 mg, 0.43 mmol). The reaction mixture was stirred at 70 °C. After 15 min, methyl-bromoacetate (15 μL, 0.16 mmol) was added dropwise. The mixture was stirred for 3 h at 70 °C. The progress of the reaction was monitored by TLC (*n*Hex:EtOAc, *v/v* 1:3). After the completion of the reaction, the mixture was cooled, and the yellow precipitate was filtered off, washed with a small amount of absolute ethanol, and dried.

*1′-{[(2Z)-3-(4-Nitrophenyl)-4-oxo-1,3-thiazolidin-2-ylidene]amino}-5′-oxo-1′,5′-dihydro-10H-spiro[acridine-9,2′-pyrrole]-4′-carbonitrile* (**7e**). Yellow solid; yield: 41% (0.022 g), m. p.: 244–247 °C (d). ^1^H NMR (600 MHz, DMSO-*d*_6_): *δ* 9.75 (s, 1H, H-10′), 8.70 (s, 1H, H-3), 8.14 (d, *J* = 9.1 Hz, 2H, H-3″,5″), 7.35 (d, *J* = 9.1 Hz, 2H, H-2″,6″), 7.26 (ddd, *J* = 8.4, 7.1, 1.4 Hz, 2H, H-3′,6′), 7.02 (dd, *J* = 8.0, 1.4 Hz, 2H, H-1′,8′), 7.02 (dd, *J* = 8.4, 1.2 Hz, 2H, H-4′,5′), 6.82 (ddd, *J* = 8.0, 7.1, 1.2 Hz, 2H, H-2′,7′), 4.08 (s, 2H, H-12). ^13^C NMR (151 MHz, DMSO-*d*_6_): *δ* 170.5 (C-11), 165.5 (C-9), 162.0 (C-3), 157.5 (C-5), 146.4 (C-4″), 139.4 (C-1″), 139.0 (C-4′a,10′a), 129.8 (C-3′,6′), 128.8 (C-2″,6″), 127.7 (C-1′,8′), 123.6 (C-3″,5″), 120.1 (C-2′,7′), 114.8 (C-4′,5′), 112.4 (C-6), 112.3 (C-8′a,9′a), 110.8 (C-4), 69.7 (C-9′), 33.3 (C-12). ^15^N NMR (61 MHz, DMSO-*d*_6_): *δ* −283.6 (N-10′), −218.5 (N-10), −186.5 (N-1), −11.6 (NO_2_). HRMS: *m*/*z* [M + H]^+^ calcd. for C_26_H_16_N_6_O_4_S: 509.10265; found: 509.10330.

### 3.12. General Synthetic Procedure for Compounds ***8a**–**e***

To a suspension of **6a**–**e** (50 mg, 0.11 mmol) in dry ethanol (0.8 mL), an equimolar amount of diethyl acetylene-dicarboxylate was added. The reaction mixture was stirred at 70 °C for 7 h. The progress of the reaction was monitored by TLC (*n*Hex:EtOAc, *v/v* 1:3). After the completion of the reaction, the mixture was cooled, and the yellow precipitate was filtered off, washed with a small amount of absolute ethanol, and dried.

*Ethyl 2-[(2Z,5Z)-2-({4′-cyano-5′-oxo-1′,5′-dihydro-10H-spiro[acridine-9,2′-pyrrol]-1′-yl}imino)-4-oxo-3-phenyl-1,3-thiazolidin-5-ylidene]acetate* (**8a**). Yellow solid; yield: 75% (0.048 g), m. p.: 256–259 °C (d). ^1^H NMR (600 MHz, DMSO-*d*_6_): *δ* 9.80 (s, 1H, H-10′), 8.77 (s, 1H, H-3), 7.33 (m, 3H, H-3″,4″,5″), 7.25 (ddd, *J* = 8.2, 7.1, 1.4 Hz, 2H, H-3′,6′), 7.15 (m, 2H, H-2″,6″), 7.07 (dd, *J* = 7.9, 1.4 Hz, 2H, H-1′,8′), 7.01 (dd, *J* = 8.2, 1.2 Hz, 2H, H-4′,5′), 6.82 (ddd, *J* = 8.0, 7.1, 1.2 Hz, 2H, H-2′,7′), 6.73 (s, 1H, H-14), 4.25 (q, *J* = 7.1 Hz, 2H, H-16), 1.26 (t, *J* = 7.1 Hz, 3H, H-17). ^13^C NMR (151 MHz, DMSO-*d*_6_): *δ* 165.2 (C-15), 163.1 (C-11), 162.1 (C-3), 161.9 (C-9), 157.8 (C-5), 140.0 (C-12), 139.1 (C-4′a,10′a), 133.0 (C-1″), 129.9 (C-3′,6′), 128.6 (C-4″), 128.5 (C-3″,5″), 127.6 (C-1′,8′), 127.4 (C-2″,6″), 120.1 (C-2′,7′), 116.3 (C-14), 114.9 (C-4′,5′), 112.3 (C-6), 112.1 (C-8′a,9′a), 111.0 (C-4), 70.1 (C-9′), 61.7 (C-16), 14.0 (C-17). ^15^N NMR (61 MHz, DMSO-*d*_6_): *δ* −283.1 (N-10′), −219.4 (N-10), −184.7 (N-1). HRMS: *m*/*z* [M + H]^+^ calcd. for C_30_H_21_N_5_O_4_S: 548.13870; found: 548.13920.

*Ethyl 2-[(2Z,5Z)-2-({4′-cyano-5′-oxo-1′,5′-dihydro-10H-spiro[acridine-9,2′-pyrrol]-1′-yl}imino)-3-(4-methoxyphenyl)-4-oxo-1,3-thiazolidin-5-ylidene]acetate* (**8b**). Yellow solid; yield: 57% (0.036 g), m. p.: 203–207 °C (d). ^1^H NMR (600 MHz, DMSO-*d*_6_): *δ* 9.78 (s, 1H, H-10′), 8.77 (s, 1H, H-3), 7.26 (ddd, *J* = 8.2, 7.1, 1.4 Hz, 2H, H-3′,6′), 7.07 (dd, *J* = 7.8, 1.4 Hz, 2H, H-1′,8′), 7.05 (d, *J* = 9.0 Hz, 2H, H-2″,6″), 7.01 (dd, *J* = 8.2, 1.2 Hz, 2H, H-4′,5′), 6.86 (d, *J* = 9.0 Hz, 2H, H-3″,5″), 6.83 (ddd, *J* = 8.1, 7.1, 1.2 Hz, 2H, H-2′,7′), 6.71 (s, 1H, H-14), 4.24 (q, *J* = 7.1 Hz, 2H, H-16), 3.75 (s, 3H, OCH_3_), 1.25 (t, *J* = 7.1 Hz, 3H, H-17). ^13^C NMR (151 MHz, DMSO-*d*_6_): *δ* 165.2 (C-15), 163.2 (C-11), 162.1 (C-3), 161.9 (C-9), 158.9 (C-4″), 157.7 (C-5), 140.0 (C-12), 139.1 (C-4′a,10′a), 129.9 (C-3′,6′), 128.5 (C-2″,6″), 127.6 (C-1′,8′), 125.5 (C-1″), 120.2 (C-2′,7′), 116.2 (C-14), 114.9 (C-4′,5′), 113.7 (C-3″,5″), 112.3 (C-6), 112.2 (C-8′a,9′a), 111.1 (C-4), 70.1 (C-9′), 61.7 (C-16), 55.4 (OCH_3_), 14.0 (C-17). ^15^N NMR (61 MHz, DMSO-*d*_6_): *δ* −282.6 (N-10′), −220.0 (N-10), −184.1 (N-1). HRMS: *m*/*z* [M + H]^+^ calcd. for C_31_H_23_N_5_O_5_S: 578.14927; found: 578.15000.

*Ethyl 2-[(2Z,5Z)-2-({4′-cyano-5′-oxo-1′,5′-dihydro-10H-spiro[acridine-9,2′-pyrrol]-1′-yl}imino)-3-(3-methoxyphenyl)-4-oxo-1,3-thiazolidin-5-ylidene]acetate* (**8c**). Yellow solid; yield: 66% (0.042 g), m. p.: 239–243 °C (d). ^1^H NMR (600 MHz, DMSO-*d*_6_): *δ* 9.73 (s, 1H, H-10′), 8.73 (s, 1H, H-3), 7.24 (ddd, *J* = 8.4, 7.1, 1.4 Hz, 2H, H-3′,6′), 7.22 (t, *J* = 8.2 Hz, 1H, H-5″), 7.06 (dd, *J* = 7.9, 1.4 Hz, 2H, H-1′,8′), 6.98 (dd, *J* = 8.2, 1.2 Hz, 2H, H-4′,5′), 6.90 (ddd, *J* = 8.4, 2.6, 0.9 Hz, 1H, H-4″), 6.82 (ddd, *J* = 8.2, 7.1, 1.2 Hz, 2H, H-2′,7′), 6.75 (t, *J* = 1.8 Hz, 1H, H-2″), 6.72 (s, 1H, H-14), 6.70 (ddd, *J* = 7.9, 2.0, 0.9 Hz, 1H, H-6″), 4.25 (q, *J* = 7.1 Hz, 2H, H-16), 3.67 (s, 3H, OCH_3_), 1.26 (t, *J* = 7.1 Hz, 3H, H-17). ^13^C NMR (151 MHz, DMSO-*d*_6_): *δ* 165.2 (C-15), 163.1 (C-11), 162.2 (C-3), 161.9 (C-9), 159.2 (C-3″), 158.0 (C-5), 140.0 (C-12), 139.1 (C-4′a,10′a), 133.9 (C-1″), 129.9 (C-3′,6′), 129.2 (C-5″), 127.6 (C-1′,8′), 120.1 (C-2′,7′), 119.6 (C-6″), 116.3 (C-14), 115.0 (C-4′,5′), 114.6 (C-4″), 113.2 (C-2″), 112.3 (C-6), 111.9 (C-8′a,9′a), 110.5 (C-4), 70.1 (C-9′), 61.7 (C-16), 55.3 (OCH_3_), 14.0 (C-17). ^15^N NMR (61 MHz, DMSO-*d*_6_): *δ* −283.5 (N-10′), −184.9 (N-1). HRMS: *m*/*z* [M + H]^+^ calcd. For C_31_H_23_N_5_O_5_S: 578.14927; found: 578.15000.

*Ethyl 2-[(2Z,5Z)-2-({4′-cyano-5′-oxo-1′,5′-dihydro-10H-spiro[acridine-9,2′-pyrrol]-1′-yl}imino)-3-(4-fluorophenyl)-4-oxo-1,3-thiazolidin-5-ylidene]acetate* (**8d**). Yellow solid; yield: 60% (0.038 g), m. p.: 269–272 °C (d). ^1^H NMR (600 MHz, DMSO-*d*_6_): *δ* 9.75 (s, 1H, H-10′), 8.76 (s, 1H, H-3), 7.26 (ddd, *J* = 8.4, 7.1, 1.4 Hz, 2H, H-3′,6′), 7.19 (dd, *J* = 9.3, 5.4 Hz, 2H, H-2″,6″), 7.16 (t, *J* = 9.3 Hz, 2H, H-3″,5″), 7.06 (dd, *J* = 8.2, 1.4 Hz, 2H, H-1′,8′), 7.00 (dd, *J* = 8.2, 1.2 Hz, 2H, H-4′,5′), 6.83 (ddd, *J* = 8.2, 7.2, 1.2 Hz, 2H, H-2′,7′), 6.72 (s, 1H, H-14), 4.24 (q, *J* = 7.1 Hz, 2H, H-16), 1.25 (t, *J* = 7.1 Hz, 3H, H-17). ^13^C NMR (151 MHz, DMSO-*d*_6_): *δ* 165.2 (C-15), 163.1 (C-11), 162.1 (C-3), 161.4 (d, *J* = 246.4 Hz, C-4″), 161.2 (C-9), 157.8 (C-5), 140.1 (C-12), 139.1 (C-4′a,10′a), 129.9 (C-3′,6′), 129.7 (d, *J* = 9.1 Hz, C-2″,6″), 129.2 (d, *J* = 2.9 Hz, C-1″), 127.6 (C-1′,8′), 120.2 (C-2′,7′), 116.3 (C-14), 115.4 (d, *J* = 23.0 Hz, H-3″,5″), 114.9 (C-4′,5′), 112.3 (C-6), 112.1 (C-8′a,9′a), 111.0 (C-4), 70.1 (C-9′), 61.7 (C-16), 14.0 (C-17). ^15^N NMR (61 MHz, DMSO-*d*_6_): *δ* −283.3 (N-10′), −184.9 (N-1). HRMS: *m*/*z* [M + H]^+^ calcd. for C_30_H_20_FN_5_O_4_S: 566.12928; found: 566.13000.

*Ethyl 2-[(2Z,5Z)-2-({4′-cyano-5′-oxo-1′,5′-dihydro-10H-spiro[acridine-9,2′-pyrrol]-1′-yl}imino)-3-(4-nitrophenyl)-4-oxo-1,3-thiazolidin-5-ylidene]acetate* (**8e**). Yellow solid; yield: 60% (0.038 g), m. p.: 183–186 °C (d). ^1^H NMR (600 MHz, DMSO-*d*_6_): *δ* 9.77 (s, 1H, H-10′), 8.76 (s, 1H, H-3), 8.18 (d, *J* = 9.1 Hz, 2H, H-3″,5″), 7.44 (d, *J* = 9.1 Hz, 2H, H-2″,6″), 7.24 (ddd, *J* = 8.2, 7.2, 1.4 Hz, 2H, H-3′,6′), 7.06 (dd, *J* = 8.0, 1.4 Hz, 2H, H-1′,8′), 6.99 (dd, *J* = 8.2, 1.2 Hz, 2H, H.4′,5′), 6.82 (ddd, *J* = 8.0, 7.2, 1.2 Hz, 2H, H-2′,7′), 6.75 (s, 1H, H-14), 4.25 (q, *J* = 7.1 Hz, 2H, H-16), 1.26 (t, *J* = 7.1 Hz, 3H, H-17). ^13^C NMR (151 MHz, DMSO-*d*_6_): *δ* 165.3 (C-15), 162.6 (C-11), 162.1 (C-3), 159.6 (C-9), 157.9 (C-5), 146.7 (C-4″), 140.3 (C-12), 139.0 (C-4′a,10′a), 138.6 (C-1″), 129.9 (C-3′,6′), 128.9 (C-2″,6″), 127.6 (C-1′,8′), 123.6 (C-3″,5″), 120.2 (C-2′,7′), 116.3 (C-14), 114.9 (C-4′,5′), 112.3 (C-6), 112.1 (C-8′a,9′a), 110.9 (C-4), 70.2 (C-9′), 61.8 (C-16), 14.0 (C-17). ^15^N NMR (61 MHz, DMSO-*d*_6_): *δ* −283.5 (N-10′), −221.3 (N-10), −184.9 (N-1), −11.8 (NO_2_). HRMS: *m*/*z* [M + H]^+^ calcd. for C_30_H_20_N_6_O_6_S: 593.12378; found: 593.12410.

### 3.13. General Synthetic Procedure for Compounds ***9a**–**e***

To a suspension of **6a**–**e** (50 mg, 0.11 mmol) in dry ethanol (0.6 mL), ethyl-bromopropionate (1.2 equiv) and triethylamine (1.2 equiv) were added. The reaction mixture was stirred at room temperature. After 1 h, another amount of triethylamine (1.2 equiv) was added. The reaction mixture was stirred for another 2–7 h. The progress of the reaction was monitored by TLC (*n*Hex:EtOAc, *v/v* 1:3). The formed yellow precipitate was filtered off, washed with a small amount of absolute ethanol, and dried.

*1′-{[(2Z)-5-Methyl-4-oxo-3-phenyl-1,3-thiazolidin-2-ylidene]amino}-5′-oxo-1′,5′-dihydro-10H-spiro[acridine-9,2′-pyrrole]-4′-carbonitrile* (**9a**). Yellow solid; yield: 77% (0.043 g), m. p.: 245–249 °C (d). ^1^H NMR (600 MHz, DMSO-*d*_6_): *δ* 9.81 (s, 1H, H-10′), 8.74 (s, 1H, H-3), 7.30 (m, 3H, H-3″,4″,5″), 7.28 (ddd, *J* = 8.4, 7.2, 1.2 Hz, 1H, H-6′), 7.26 (ddd, *J* = 8.4, 7.2, 1.2 Hz, 1H, H-3′), 7.07. (m, 2H, H-2″,6″), 7.04 (dd, *J* = 8.0, 1.2 Hz, 1H, H-8′), 7.03 (dd, *J* = 8.4, 1.2 Hz, 2H, H-4′,5′), 7.02 (dd, *J* = 8.0, 1.2 Hz, 1H, H-1′), 6.84 (ddd, *J* = 8.0, 7.2, 1.2 Hz, 1H, H-7′), 6.81 (ddd, *J* = 8.0, 7.2, 1.2 Hz, 1H, H-2′), 4.31 (q, *J* = 7.2 Hz, 1H, H-12), 1.33 (d, *J* = 7.2 Hz, 3H, H-14). ^13^C NMR (151 MHz, DMSO-*d*_6_): *δ* 174.24 (C-11), 166.51 (C-9), 161.84 (C-3), 156.98 (C-5), 139.27 (C-10′a), 139.19 (C-4′a), 133.88 (C-1″), 129.79 (C-6′), 129.73 (C-3′), 128.49 (C-3″,5″), 128.32 (C-4″), 127.96 (C-8′), 127.54 (C-2″,6″), 127.49 (C-1′), 120.05 (C-7′), 119.99 (C-2′), 114.78 (C-5′), 114.62 (C-4′), 112.57 (C-9′a), 112.48 (C-8′a), 112.44 (C-6), 111.47 (C-4), 69.67 (C-9′), 42.00 (C-12), 18.77 (C-14). ^15^N NMR (61 MHz, DMSO-*d*_6_): *δ* −282.5 (N-10′), −218.3 (N-10), −185.1 (N-1). HRMS: *m*/*z* [M + H]^+^ calcd. for C_27_H_19_N_5_O_2_S: 478.13322; found: 478.13410.

*1′-{[(2Z)-3-(4-Methoxyphenyl)-5-methyl-4-oxo-1,3-thiazolidin-2-ylidene]amino}-5′-oxo-1′,5′-dihydro-10H-spiro[acridine-9,2′-pyrrole]-4′-carbonitrile* (**9b**). Yellow solid; yield: 64% (0.036 g), m. p.: 219–224 °C (d). ^1^H NMR (600 MHz, DMSO-*d*_6_): *δ* 9.78 (s, 1H, H-10′), 8.73 (s, 1H, H-3), 7.28 (ddd, *J* = 8.4, 7.1, 1.4 Hz, 1H, H-6′), 7.28 (ddd, *J* = 8.4, 7.1, 1.4 Hz, 1H, H-3′), 7.05 (dd, *J* = 8.2, 1.2 Hz, 1H, H-8′), 7.03 (dd, *J* = 8.4, 1.2 Hz, 1H, H-5′), 7.03 (dd, *J* = 8.4, 1.2 Hz, 1H, H-4′), 7.02 (dd, *J* = 8.4, 1.2 Hz, 1H, H-1′), 6.97 (d, *J* = 9.0 Hz, 2H, H-2″,6″), 6.84 (ddd, *J* = 8.4, 7.1, 1.3 Hz, 1H, H-7′), 6.83 (ddd, *J* = 8.4, 7.1, 1.3 Hz, 1H, H-2′), 6.83 (d, *J* = 9.0 Hz, 2H, H-3″,5″), 4.28 (q, *J* = 7.2 Hz, 1H, H-12), 3.73 (s, 3H, OCH_3_), 1.32 (d, *J* = 7.2 Hz, 3H, H-14). ^13^C NMR (151 MHz, DMSO-*d*_6_): *δ* 174.33 (C-11), 166.64 (C-9), 161.82 (C-3), 158.71 (C-4″), 156.93 (C-5), 139.27 (C-10′a), 139.19 (C-4′a), 129.79 (C-6′), 129.75 (C-3′), 128.67 (C-2″,6″), 127.96 (C-8′), 127.49 (C-1′), 126.48 (C-1″), 120.07 (C-7′), 120.00 (C-2′), 114.79 (C-5′), 114.63 (C-4′), 113.68 (C-3″,5″), 112.61 (C-9′a), 112.53 (C-8′a), 112.46 (C-6), 111.55 (C-4), 69.67 (C-9′), 55.33 (OCH_3_), 41.88 (C-12), 18.76 (C-14). ^15^N NMR (61 MHz, DMSO-*d*_6_): *δ* −283.3 (N-10′), −186.0 (N-1). HRMS: *m*/*z* [M + H]^+^ calcd. for C_28_H_21_N_5_O_3_S: 508.14379; found: 508.14420.

*1′-{[(2Z)-3-(3-Methoxyphenyl)-5-methyl-4-oxo-1,3-thiazolidin-2-ylidene]amino}-5′-oxo-1′,5′-dihydro-10H-spiro[acridine-9,2′-pyrrole]-4′-carbonitrile* (**9c**). Yellow solid; yield: 64% (0.037 g), m. p.: 237–240 °C (d). ^1^H NMR (600 MHz, DMSO-*d*_6_): *δ* 9.74 (s, 1H, H-10′), 8.70 (s, 1H, H-3), 7.27 (ddd, *J* = 8.4, 7.2, 1.2 Hz, 1H, H-6′), 7.25 (ddd, *J* = 8.4, 7.2, 1.2 Hz, 1H, H-3′), 7.20 (t, *J* = 8.1 Hz, 1H, H-5″), 7.04 (dd, *J* = 8.0, 1.2 Hz, 1H, H-8′), 7.03 (dd, *J* = 8.0, 1.2 Hz, 1H, H-1′), 7.01 (dd, *J* = 8.4, 1.2 Hz, 2H, H-4′,5′), 6.87 (ddd, *J* = 8.4, 2.5, 0.9 Hz, 1H, H-4″), 6.83 (ddd, *J* = 8.0, 7.2, 1.2 Hz, 1H, H-7′), 6.81 (ddd, *J* = 8.0, 7.2, 1.2 Hz, 1H, H-2′), 6.65 (t, *J* = 2.2 Hz, 1H, H-2″), 6.63 (ddd, *J* = 7.8, 1.9, 0.9 Hz, 1H, H-6″), 4.30 (q, *J* = 7.2 Hz, 1H, H-12), 3.67 (s, 3H, OCH_3_), 1.34 (d, *J* = 7.2 Hz, 3H, H-14). ^13^C NMR (151 MHz, DMSO-*d*_6_): *δ* 174.15 (C-11), 166.33 (C-9), 161.91 (C-3), 159.22 (C-3″), 157.12 (C-5), 139.23 (C-10′a), 139.20 (C-4′a), 134.89 (C-1″), 129.74 (C-6′), 129.69 (C-3′), 129.16 (C-5″), 127.95 (C-8′), 127.53 (C-1′), 120.00 (C-7′), 119.93 (C-2′), 119.79 (C-6″), 114.87 (C-5′), 114.71 (C-4′), 114.21 (C-4″), 113.29 (C-2″), 112.45 (C-6), 112.38 (C-9′a), 112.29 (C-8′a), 110.89 (C-4), 69.65 (C-9′), 55.28 (OCH_3_), 41.98 (C-12), 18.70 (C-14). ^15^N NMR (61 MHz, DMSO-*d*_6_): *δ* −283.3 (N-10′), −186.0 (N-1). HRMS: *m*/*z* [M + H]^+^ calcd. for C_28_H_21_N_5_O_3_S: 508.14379; found: 508.14450.

*1′-{[(2Z)-3-(4-Fluorophenyl)-5-methyl-4-oxo-1,3-thiazolidin-2-ylidene]amino}-5′-oxo-1′,5′-dihydro-10H-spiro[acridine-9,2′-pyrrole]-4′-carbonitrile* (**9d**). Yellow solid; yield: 73% (0.041 g), m. p.: 235–239 °C (d). ^1^H NMR (600 MHz, DMSO-*d*_6_): *δ* 9.77 (s, 1H, H-10′), 8.74 (s, 1H, H-3′), 7.28 (ddd, *J* = 8.4, 7.1, 1.4 Hz, 1H, H-6′), 7.28 (ddd, *J* = 8.4, 7.1, 1.4 Hz, 1H, H-3′), 7.13 (m, 4H, H-2″,3″,5″,6″), 7.03 (m, 4H, H-1′,4′,5′,8′), 6.84 (ddd, *J* = 8.1, 7.2, 1.3 Hz, 1H, H-7′), 6.82 (ddd, *J* = 8.1, 7.1, 1.3 Hz, 1H, H-2′), 4.30 (q, *J* = 7.2 Hz, 1H, H-12), 1.33 (d, *J* = 7.2 Hz, 2H, H-14). ^13^C NMR (151 MHz, DMSO-*d*_6_): *δ* 174.17 (C-11), 166.20 (C-9), 161.85 (C-3), 161.25 (d, *J* = 245.8 Hz, C-4″), 157.04 (C-5), 139.21 (C-10′a), 139.15 (C-4′a), 130.9 (d, *J* = 2.9 Hz, C-1″), 129.81 (C-6′), 129.80 (d, *J* = 9.0 Hz, C-2″,6″), 129.79 (C-3′), 127.91 (C-8′), 127.48 (C-1′), 120.10 (C-7′), 120.04 (C-2′), 115.40 (d, *J* = 22.8 Hz, C-3″,5″), 114.79 (C-5′), 114.64 (C-4′), 112.59 (C-9′a), 112.51 (C-8′a), 112.44 (C-6), 111.49 (C-4), 69.69 (C-9′), 42.12 (C-12), 18.68 (C-14). ^15^N NMR (61 MHz, DMSO-*d*_6_): *δ* −282.6 (N-10′), −220.1 (N-10), −185.5 (N-1). HRMS: *m*/*z* [M + H]^+^ calcd. for C_27_H_18_FN_5_O_2_S: 496.1238; found: 496.12470.

*1′-{[(2Z)-5-Methyl-3-(4-nitrophenyl)-4-oxo-1,3-thiazolidin-2-ylidene]amino}-5′-oxo-1′,5′-dihydro-10H-spiro[acridine-9,2′-pyrrole]-4′-carbonitrile* (**9e**). Yellow solid; yield: 38% (0.021 g), m. p.: 170–173 °C (d). ^1^H NMR (600 MHz, DMSO-*d*_6_): *δ* 9.79 (s, 1H, H-10′), 8.73 (s, 1H, H-3), 8.16 (d, *J* = 8.8 Hz, 2H, H-3″,5″), 7.38 (d, *J* = 8.8 Hz, 2H, H-2″,6″), 7.28 (ddd, *J* = 8.4, 7.2, 1.2 Hz, 1H, H-6′), 7.27 (ddd, *J* = 8.4, 7.2, 1.2 Hz, 1H, H-3′), 7.04 (m, 4H, H-1′,4′,5′,8′), 6.84 (ddd, *J* = 8.4, 7.1, 1.5 Hz, 1H, H-7′), 6.82 (ddd, *J* = 8.4, 7.1, 1.5 Hz, 1H, H-2′), 4.37 (q, *J* = 7.2 Hz, 1H, H-12), 1.37 (d, *J* = 7.2 Hz, 3H, H-14). ^13^C NMR (151 MHz, DMSO-*d*_6_): *δ* 173.77 (C-11), 165.06 (C-9), 161.93 (C-3), 157.18 (C-5), 146.57 (C-4″), 139.39 (C-1″), 139.13 (C-10′a), 139.11 (C-4′a), 129.86 (C-3′,6′), 129.00 (C-2″,6″), 127.87 (C-8′), 127.58 (C-1′), 123.68 (C-3″,5″), 120.21 (C-7′), 120.13 (C-2′), 114.85 (C-5′), 114.72 (C-4′), 112.53 (C-9′a), 112.46 (C-8′a), 112.42 (C-6), 111.32 (C-4), 69.77 (C-9′), 42.62 (C-12), 18.52 (C-14). ^15^N NMR (61 MHz, DMSO-*d*_6_): *δ* −283.3 (N-10′), −220.7 (N-10), −186.1 (N-1), −11.6 (NO_2_). HRMS: *m*/*z* [M + H]^+^ calcd. for C_27_H_18_N_6_O_4_S: 523.1183; found: 523.11880.

### 3.14. General Synthetic Procedure for Compounds ***10a**–**e***

To a suspension of **6a**–**e** (50 mg, 0.11 mmol) in dry ethanol (0.6 mL), ethyl-2-bromovalerate (2.0 equiv) and triethylamine (2.0 equiv) were added. The reaction mixture was stirred at room temperature. After 1.5 h, a second amount of triethylamine (2.0 equiv) was added. The mixture was stirred at room temperature for 24–48 h. After the completion of the reaction, the yellow precipitate formed was filtered off, washed with a small amount of absolute ethanol, and dried.

*5′-Oxo-1′-{[(2Z)-4-oxo-3-phenyl-5-propyl-1,3-thiazolidin-2-ylidene]amino}-1′,5′-dihydro-10H-spiro[acridine-9,2′-pyrrole]-4′-carbonitrile* (**10a**). Yellow solid; yield: 69% (0.041 g), m. p.: 238–240 °C (d). ^1^H NMR (600 MHz, DMSO-*d*_6_): *δ* 9.83 (s, 1H, H-10′), 8.75 (s, 1H, H-3), 7.31 (m, 3H, H-3″,4″,5″), 7.28 (ddd, *J* = 8.3, 7.1, 1.4 Hz, 1H, H-6′), 7.25 (ddd, *J* = 8.3, 7.1, 1.4 Hz, 1H, H-3′), 7.13 (dd, *J* = 8.0, 1.4 Hz, 1H, H-8′), 7.04 (m, 4H, H-4′,5′,2″,6″), 6.97 (dd, *J* = 8.0, 1.4 Hz, 1H, H-1′), 6.86 (ddd, *J* = 8.2, 7.2, 1.2 Hz, 1H, H-7′), 6.79 (ddd, *J* = 8.1, 7.1, 1.2 Hz, 1H, H-2′), 4.40 (dd, *J* = 7.2, 4.2 Hz, 1H, H-12), 1.72 (m, 1H, H-14), 1.56 (m, 1H, H-14), 1.08 (m, 1H, H-15), 0.94 (m, 1H, H-15), 0.79 (t, *J* = 7.3 Hz, 3H, H-16). ^13^C NMR (151 MHz, DMSO-*d*_6_): *δ* 173.57 (C-11), 167.00 (C-9), 161.78 (C-3), 156.76 (C-5), 139.44 (C-4′a), 139.14 (C-10′a), 133.78 (C-1″), 129.74 (C-6′), 129.66 (C-3′), 128.54 (C-3″,5″), 128.35 (C-4″), 128.25 (C-8′), 127.50 (C-2″,6″), 127.31 (C-1′), 120.05 (C-7′), 119.92 (C-2′), 114.77 (C-4′), 114.59 (C-5′), 112.59 (C-9′a), 112.46 (C-6), 112.44 (C-8′a), 111.75 (C-4), 69.64 (C-9′), 47.45 (C- 12), 34.04 (C-14), 17.76 (C-15), 13.34 (C-16). ^15^N NMR (61 MHz, DMSO-*d*_6_): *δ* −283.0 (N-10′), −185.2 (N-1). HRMS: *m*/*z* [M + H]^+^ calcd. for C_29_H_23_N_5_O_2_S: 506.16452; found: 506.16510.

*1′-{[(2Z)-3-(4-Methoxyphenyl)-4-oxo-5-propyl-1,3-thiazolidin-2-ylidene]amino}-5′-oxo-1′,5′-dihydro-10H-spiro[acridine-9,2′-pyrrole]-4′-carbonitrile* (**10b**). Yellow solid; yield: 46% (0.027 g), m. p.: 247–250 °C (d). ^1^H NMR (600 MHz, DMSO-*d*_6_): *δ* 9.80 (s, 1H, H-10′), 8.75 (s, 1H, H-3), 7.29 (ddd, *J* = 8.4, 7.2, 1.4 Hz, 1H, H-6′), 7.27 (ddd, *J* = 8.4, 7.2, 1.4 Hz, 1H, H-3′), 7.13 (dd, *J* = 8.0, 1.4 Hz, 1H, H-8′), 7.05 (dd, *J* = 8.0, 1.2 Hz, 1H, H-4′), 7.03 (dd, *J* = 8.0, 1.2 Hz, 1H, H-5′), 6.97 (dd, *J* = 8.0, 1.4 Hz, 1H, H-1′), 6.95 (d, *J* = 9.0 Hz, 2H, H-2″,6″), 6.86 (ddd, *J* = 8.0, 7.2, 1.2 Hz, 1H, H-7′), 6.84 (d, *J* = 9.0 Hz, H-3″,5″), 6.80 (ddd, *J* = 8.0, 7.2, 1.2 Hz, 1H, H-2′), 4.36 (dd, *J* = 7.2, 4.2 Hz, 1H, H-12), 3.73 (s, 3H, OCH_3_), 1.71 (m, 1H, H-14), 1.54 (m, 1H, H-14), 1.06 (m, 1H, H-15), 0.91 (m, 1H, H-15), 0.79 (t, *J* = 7.3 Hz, 3H, H-16). ^13^C NMR (151 MHz, DMSO-*d*_6_): *δ* 173.68 (C-11), 167.14 (C-9), 161.76 (C-3), 158.76 (C-4″), 156.72 (C-5), 139.45 (C-4′a), 139.15 (C-10′a), 129.75 (C-6′), 129.70 (C-3′), 128.65 (C-2″,6″), 128.25 (C-8′), 127.31 (C-1′), 126.39 (C-1″), 120.09 (C-2′), 119.95 (C-7′), 114.81 (C-4′), 114.61 (C-5′), 113.75 (C-3″,5″), 112.64 (C-9′a), 112.49 (C-6), 112.49 (C-8′a), 111.84 (C-4), 69.64 (C-9′), 55.35 (OCH_3_), 47.34 (C-12), 34.05 (C-14), 17.79 (C-15), 13.35 (C-16). ^15^N NMR (61 MHz, DMSO-*d*_6_): *δ* −283.0 (N-10′), −218.5 (N-10), −185.0 (N-1). HRMS: *m*/*z* [M + H]^+^ calcd. for C_30_H_25_N_5_O_3_S: 536.17509; found: 536.17560.

*1′-{[(2Z)-3-(3-Methoxyphenyl)-4-oxo-5-propyl-1,3-thiazolidin-2-ylidene]amino}-5′-oxo-1′,5′-dihydro-10H-spiro[acridine-9,2′-pyrrole]-4′-carbonitrile* (**10c**). Yellow solid; yield: 71% (0.042 g), m. p.: 229–231 °C (d). ^1^H NMR (600 MHz, DMSO-*d*_6_): *δ* 9.76 (s, 1H, H-10′), 8.72 (s, 1H, H-3), 7.28 (ddd, *J* = 8.4, 7.1, 1.4 Hz, 1H, H-6′), 7.24 (ddd, *J* = 8.4, 7.1, 1.4 Hz, 1H, H-3′), 7.21 (t, *J* = 8.1 Hz, 1H, H-5″), 7.12 (dd, *J* = 8.0, 1.4 Hz, 1H, H-8′), 7.02 (dd, *J* = 8.2, 1.2 Hz, 1H, H-4′), 7.01 (dd, *J* = 8.2, 1.2 Hz, 1H, H-5′), 6.98 (dd, *J* = 8.0, 1.4 Hz, 1H, H-1′), 6.88 (ddd, *J* = 8.1, 2.6, 1.0 Hz, 1H, H-4″), 6.85 (ddd, *J* = 8.4, 7.2, 1.2 Hz, 1H, H-7′), 6.80 (ddd, *J* = 8.2, 7.2, 1.2 Hz, 1H, H-2′), 6.62 (t, *J* = 2.0 Hz, 1H, H-2″), 6.60 (ddd, *J* = 7.8, 2.0, 1.0 Hz, 1H, H-6″), 4.38 (dd, *J* = 7.2, 4.2 Hz, 1H, H-12), 3.68 (s, 3H, OCH_3_), 1.72 (m, 1H, H-14), 1.57 (m, 1H, H-14), 1.08 (m, 1H, H-15), 0.95 (m, 1H, H-15), 0.80 (t, *J* = 7.3 Hz, 3H, H-16). ^13^C NMR (151 MHz, DMSO-*d*_6_): *δ* 173.52 (C-11), 166.93 (C-9), 161.87 (C-3), 159.28 (C-3″), 156.89 (C-5), 139.43 (C-4′a), 139.19 (C-10′a), 134.79 (C-1″), 129.73 (C-6′), 129.63 (C-3′), 129.26 (C-5″), 128.28 (C-8′), 127.35 (C-1′), 120.03 (C-7′), 119.88 (C-2′), 119.78 (C-6″), 114.90 (C-4′), 114.68 (C-5′), 114.14 (C-4″), 113.41 (C-2″), 112.49 (C-6), 112.44 (C-9′a), 112.28 (C-8′a), 111.31 (C-4), 69.62 (C-9′), 55.32 (OCH_3_), 47.44 (C-12), 34.08 (C-14), 17.86 (C-15), 13.37 (C-16). ^15^N NMR (61 MHz, DMSO-*d*_6_): *δ* −282.6 (N-10′), −184.8 (N-1). HRMS: *m*/*z* [M + H]^+^ calcd. for C_30_H_25_N_5_O_3_S: 536.17509; found: 536.17570.

*1′-{[(2Z)-3-(4-Fluorophenyl)-4-oxo-5-propyl-1,3-thiazolidin-2-ylidene]amino}-5′-oxo-1′,5′-dihydro-10H-spiro[acridine-9,2′-pyrrole]-4′-carbonitrile* (**10d**). Yellow solid; yield: 46% (0.026 g), m. p.: 230–232 °C (d). ^1^H NMR (600 MHz, DMSO-*d*_6_): *δ* 9.79 (s, 1H, H-10′), 8.75 (s, 1H, H-3), 7.29 (ddd, *J* = 8.4, 7.2, 1.8 Hz, 1H, H-6′), 7.27 (ddd, *J* = 8.4, 7.2, 1.8 Hz, 1H, H-3′), 7.14 (m, 3H, H-8′,3″,5″), 7.09 (m, 2H, H-2″,6″), 7.03 (t, *J* = 7.6 Hz, 2H, H-4′,5′), 6.98 (dd, *J* = 8.4, 1.8 Hz, 1H, H-1′), 6.86 (ddd, *J* = 8.4, 7.2, 1.2 Hz, 1H, H-7′), 6.80 (ddd, *J* = 8.4, 7.2, 1.2 Hz, 1H, H-2′), 4.39 (dd, *J* = 7.1, 4.2 Hz, 1H, H-12), 1.72 (m, 1H, H-14), 1.57 (m, 1H, H-14), 1.07 (m, 1H, H-15), 0.93 (m, 1H, H-15), 0.79 (t, *J* = 7.3 Hz, 3H, H-16). ^13^C NMR (151 MHz, DMSO-*d*_6_): *δ* 173.53 (C-11), 166.75 (C-9), 161.79 (C-3), 161.29 (d, *J* = 245.8 Hz, C-4″), 156.80 (C-5), 139.40 (C-4′a), 139.10 (C-10′a), 129.98 (d, *J* = 2.8 Hz, C-1″), 129.79 (C-6′), 129.75 (d, *J* = 6.8 Hz, C-2″,6″), 129.74 (C-3′), 128.21 (C-8′), 127.31 (C-1′), 120.10 (C-7′), 120.00 (C-2′), 115.48 (d, *J* = 23.0 Hz, C-3″,5″), 114.79 (C-4′), 114.61 (C-5′), 112.63 (C-9′a), 112.50 (C-8′a), 112.46 (C-6), 111.82 (C-4), 69.65 (C-9′), 47.56 (C-12), 33.97 (C-14), 17.79 (C-15), 13.36 (C-16). ^15^N NMR (61 MHz, DMSO-*d*_6_): *δ* −283.0 (N-10′), −219.2 (N-10), −185.5 (N-1). HRMS: *m*/*z* [M + H]^+^ calcd. for C_29_H_22_FN_5_O_2_S: 524.1551; found: 524.15610.

*1′-{[(2Z)-3-(4-Nitrophenyl)-4-oxo-5-propyl-1,3-thiazolidin-2-ylidene]amino}-5′-oxo-1′,5′-dihydro-10H-spiro[acridine-9,2′-pyrrole]-4′-carbonitrile* (**10e**). Yellow solid; yield: 60% (0.035 g), m. p.: 249–251 °C (d). ^1^H NMR (600 MHz, DMSO-*d*_6_): *δ* 9.82 (s, 1H, H-10′), 8.76 (s, 1H, H-3), 8.17 (d, *J* = 8.5 Hz, 2H, H-3″,5″), 7.35 (d, *J* = 8.5 Hz, 2H, H-2″,6″), 7.30 (ddd, *J* = 8.4, 7.2, 1.2 Hz, 1H, H-6′), 7.27 (ddd, *J* = 8.4, 7.2, 1.2 Hz, 1H, H-3′), 7.13 (d, *J* = 8.4 Hz, 1H, H-8′), 7.06 (d, *J* = 8.1 Hz, 1H, H-4′), 7.03 (d, *J* = 8.1 Hz, 1H, H-5′), 6.99 (d, *J* = 8.4 Hz, 1H, H-1′), 6.87 (ddd, *J* = 8.4, 7.2, 1.2 Hz, 1H, H-7′), 6.81 (ddd, *J* = 8.4, 7.2, 1.3 Hz, 1H, H-2′), 4.45 (dd, *J* = 7.2, 4.2 Hz, 1H, H-12), 1.75 (m, 1H, H-14), 1.62 (m, 1H, H-14), 1.10 (m, 1H, H-15), 0.95 (m, 1H, H-15), 0.79 (t, *J* = 7.3 Hz, 3H, H-16). ^13^C NMR (151 MHz, DMSO-*d*_6_): *δ* 173.12 (C-11), 165.82 (C-9), 161.80 (C-3), 156.85 (C-5), 146.64 (C-4″), 139.29 (C-4′a), 139.20 (C-10′a), 139.03 (C-1″), 129.82 (C-3′,6′), 128.94 (C-2″,6″), 128.16 (C-8′), 127.39 (C-1′), 123.75 (C-3″,5″), 120.22 (C-2′), 120.08 (C-7′), 114.80 (C-4′), 114.67 (C-5′), 112.57 (C-9′a), 112.44 (C-8′a), 112.41 (C-6), 111.70 (C-4), 69.70 (C-9′), 47.99 (C-12), 33.89 (C-14), 17.85 (C-15), 13.36 (C-16). ^15^N NMR (61 MHz, DMSO-*d*_6_): *δ* −283.2 (N-10′), −219.3 (N-10), −185.7 (N-1), −11.7 (NO_2_). HRMS: *m*/*z* [M + H]^+^ calcd. for C_29_H_22_N_6_O_4_S: 551.1496; found: 551.115050.

## 4. Conclusions

In this study, we showcased the potential of novel derivatives incorporating acridine, pyrrole, and thiazolidine rings as promising contenders in the realm of antitumor drug discovery. The strategic integration of these three distinct structural motifs paves the way for the development of potent and selective anticancer agents. The acridine ring, imparting DNA intercalation properties, synergistically combines with the anti-inflammatory attributes of the pyrrole ring and the antidiabetic characteristics of the thiazolidine ring. This intricate amalgamation offers the advantage of targeting multiple pathways associated with tumor growth and progression.

We efficiently synthesized spiro derivatives **6a**–**e** via a concise two-step process, leveraging isothiocyanates **1a**–**e** and 2-cyanoacetohydrazide as starting materials. The incorporation of the thiourea side chain proved pivotal in orchestrating regioselective reactions with bifunctional reagents, resulting in the construction of the thiazolidine-4-one ring. The successful formation of single regioisomeric products for each derivative validated the precision of our synthetic approach. The thorough characterization of the synthesized derivatives **6**–**10** was accomplished through advanced spectroscopic techniques, including 1D, 2D NMR, FT-IR, HRMS, and single-crystal structure analysis, ensuring their structural integrity.

Importantly, our investigation into the impact of the spiro derivatives **6**–**10** on various cancer cell lines unveiled compelling anticancer potential. Notably, the derivatives **8a**–**e**, featuring ester functional groups, exhibited a remarkable activity against all tested cancer cell lines, with IC_50_ values below 10 μM. Particularly, the derivatives **6c** and **6e** demonstrated exceptional selectivity and potency against HCT-116 and Jurkat cell lines. Furthermore, the derivatives **7a**–**e** showcased significant efficacy against MCF-7 and HCT-116 cancer cell lines.

These findings underscore the successful synthesis, meticulous characterization, and potent anticancer attributes of the novel spiro derivatives. Particularly promising is their potential use as lead compounds targeting colorectal cancer, leukemia, cervical adenocarcinoma, and breast adenocarcinoma. Moving forward, continued investigations and the optimization of these derivatives hold substantial promise for the development of innovative therapeutic agents to combat various cancer types.

In conclusion, our study provides valuable insights into the design and synthesis of multifaceted spiro derivatives with exceptional anticancer potential. This research sets the stage for further exploration and refinement, heralding new avenues for the creation of next-generation therapeutic interventions across diverse cancer scenarios.

## Data Availability

The data presented in this study and associated additional data are available upon request.

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
