# Peer review of "Derivatives Incorporating Acridine, Pyrrole, and Thiazolidine Rings as Promising Antitumor Agents"

_molecules, 2023, doi:10.3390/molecules28186616_

Round 1

Reviewer 1 Report

1. The article discussed very efficiently the synthesis ant its characterization.

2. Describe each and every steps of synthesis mentioned in scheme 3a-e, 5a-e and 6a-e.

3. The numbering of the compounds 4a-e is missing in numbering. it should be included in manuscript.

4. Describe structure activity relation of a series of compounds.

5. Describe the possible mechanism involved in anticancer activity if possible.

6. Include future prospects of the synthesised compound in the development of new lead compounds.

7. More recent reference must be included in the manuscript.

8. some grammatical mistakes are there in manuscript it should be removed.

Some grammatical mistakes are there in manuscript it should be removed.

Author Response

We would like to thank the reviewer for his/her comments, which have given us the opportunity to improve the manuscript.

Reviewer 1:

  1. The article discussed very efficiently the synthesis and its characterization.
  2. Describe each and every steps of synthesis mentioned in scheme 3a-e, 5a-e and 6a-e.

Thank you for your valuable feedback and your interest in the details of the synthesis steps in Scheme 1. We appreciate the opportunity to provide a comprehensive description of each step in the synthesis process of these compounds.

  1. The numbering of the compounds 4a-e is missing in numbering. it should be included in manuscript.

In the synthesis, we used only one derivative 4. It was acridine-9-carbaldehyde. Therefore there is no need to designate it as 4ae.

  1. Describe structure activity relation of a series of compounds.

The structure-activity relationship was included.

  1. Describe the possible mechanism involved in anticancer activity if possible.

As of the current stage of our study, we have not yet identified a specific mechanism responsible for the observed anticancer activity of our compounds. Our research has primarily focused on the synthesis, characterization, and evaluation of these compounds for their biological effects. While we have obtained promising results in terms of their antiproliferative effects on cancer cell lines, further investigations are needed to elucidate the precise molecular mechanisms involved.

We acknowledge the importance of understanding the mechanism of action, as it is crucial for advancing our understanding of these compounds and potentially optimizing their therapeutic potential. Our future research plans include in-depth mechanistic studies, such as cellular assays, target identification, and pathway analyses, to provide a comprehensive insight into how our compounds exert their anticancer effects.

We are committed to providing a thorough mechanistic explanation in our subsequent publications as we make progress in this area. We understand the significance of this aspect and aim to contribute valuable insights to the scientific community.

  1. Include future prospects of the synthesised compound in the development of new lead compounds.

We plan to optimize lead compound structure based on SAR analysis, identify specific molecular targets, and conduct preclinical studies. These efforts aim to develop the compound into a promising lead candidate for potential therapeutic applications.

  1. More recent reference must be included in the manuscript.

            References were included.

  1. Some grammatical mistakes are there in manuscript it should be removed.

            Grammatical mistakes were corrected.

Reviewer 2 Report

The submitted paper describes the synthesis and characterization of derivatives Incorporating acridine, pyrrole and thiazolidine ring cores, as well as their evaluation as potential antitumor agents. The paper structure is scientifically sound and experimental data are in accordance with the Author’s described results.

However, some issues need addressing. 

- line 18: “paper” instead of “report”

- line 22: “starting from” instead of “from starting”

- line 32: remove “of the phenyl ring”

- lines 72-77: these sentences should be removed since they restate what has already been describeed at the start of the page

- Scheme 1: the caption should contain: a) yields or yield ranges; b) molecules and intermediate numbers or number ranges

- line 137: “feature” instead of “features”

- lines 137-139: please provide references

- lines 178-179: please provide references

- line 180: “room temperature” instead of “laboratory temperature with”

- line 194: “explored the use” instead of “explored to use”

- lines 213-227: why has 8 been excluded by the paragraph?

- paragraph 2.6 (pg. 11-13): as MS spectra are described in detail, their presence is crucial for understanding and they should be added to the Supplementary Material

- line 376: “17 μM” instead of “13 μM

overall language is correct and does not impede reading

Author Response

We would like to thank the reviewer for his/her comments, which have given us the opportunity to improve the manuscript.

The submitted paper describes the synthesis and characterization of derivatives Incorporating acridine, pyrrole and thiazolidine ring cores, as well as their evaluation as potential antitumor agents. The paper structure is scientifically sound and experimental data are in accordance with the Author’s described results.

However, some issues need addressing. 

- line 18: “paper” instead of “report”

- line 22: “starting from” instead of “from starting”

- line 32: remove “of the phenyl ring”

- lines 72-77: these sentences should be removed since they restate what has already been describeed at the start of the page – the sentences were removed

- Scheme 1: the caption should contain: a) yields or yield ranges; b) molecules and intermediate numbers or number ranges

- line 137: “feature” instead of “features”

- lines 137-139: please provide references

- lines 178-179: please provide references

- line 180: “room temperature” instead of “laboratory temperature with”

- line 194: “explored the use” instead of “explored to use”

- line 376: “17 μM” instead of “13 μM

Thank you for your thorough and constructive review of our manuscript. We greatly appreciate your valuable feedback and are pleased to inform you that all of your suggestions have been carefully incorporated into the revised version of the article.

- lines 213-227: why has 8 been excluded by the paragraph?

During the synthesis of derivatives 8, no new asymmetric center was formed at carbon C-12. Consequently, we excluded these derivatives from the HPLC analysis.

- paragraph 2.6 (pg. 11-13): as MS spectra are described in detail, their presence is crucial for understanding and they should be added to the Supplementary Material

MS2 spectra were included in ESI.

Reviewer 3 Report

The authors showcased the potential of novel derivatives incorporating acridine, pyrrole, and thiazolidine rings as promising contenders in the realm of antitumor drug discovery. The strategic integration of these three distinct structural motifs has paved the way for the development of potent and selective anticancer agents. The acridine ring, imparting DNA intercalation properties, synergistically combines with the anti-inflammatory attributes of the pyrrole ring and the antidiabetic characteristics of the thiazolidine ring. This intricate amalgamation offers the advantage of targeting multiple pathways associated with tumor growth and progression. In my opinion, this result is interesting and would be of interest to the broad readership of Molecules. I support its publication in Molecules after addressing the following issues.

Figure 12 requires further editing as it is not entirely visible in the manuscript.

Author Response

We would like to thank the reviewer for his/her comments, which have given us the opportunity to improve the manuscript.

Reviewer 3:

The authors showcased the potential of novel derivatives incorporating acridine, pyrrole, and thiazolidine rings as promising contenders in the realm of antitumor drug discovery. The strategic integration of these three distinct structural motifs has paved the way for the development of potent and selective anticancer agents. The acridine ring, imparting DNA intercalation properties, synergistically combines with the anti-inflammatory attributes of the pyrrole ring and the antidiabetic characteristics of the thiazolidine ring. This intricate amalgamation offers the advantage of targeting multiple pathways associated with tumor growth and progression. In my opinion, this result is interesting and would be of interest to the broad readership of Molecules. I support its publication in Molecules after addressing the following issues.

Figure 12 requires further editing as it is not entirely visible in the manuscript.

I have adjusted the placement of Figure 12 to the left to ensure that the entire figure is now visible in the manuscript. I trust this adjustment resolves the visibility issue.

Round 2

Reviewer 1 Report

Accepted in present form

Reviewer 2 Report

The paper may be accepted as it is now